

# A 12-Year Long Global Record of Optical Depth of Absorbing Aerosols above the Clouds Derived from OMI/OMACA Algorithm

Hiren Jethva[1,2], Omar Torres[2], Changwoo Ahn[3]

[1]Universities Space Research Association, 7178 Columbia Gateway Drive, Columbia, MD 21046, USA
[2]NASA Goddard Space Flight Center, Earth Science Division, Code 614, Greenbelt, MD 20771, USA
[3]Science Systems and Applications, Inc., 10210 Greenbelt Rd, Lanham, MD 20706 USA

*Correspondence to*: Hiren Jethva (hiren.t.jethva@nasa.gov)

**Abstract.** Aerosol-cloud interaction continues to be one of the leading uncertain components of the climate models, primarily due to the lack of adequate knowledge of the complex microphysical and radiative processes of the aerosol-cloud system. Situations when the light-absorbing aerosols such as carbonaceous particles and windblown dust overlay low-level cloud decks are commonly found in several regions of the world. Contrary to the known cooling effects of these aerosols in cloud-free scenario over darker surfaces, an overlapping situation of the absorbing aerosols over the cloud can lead to a significant level of atmospheric absorption exerting a positive radiative forcing (warming) at the top-of-atmosphere. We contribute to this topic by introducing a new global product of the above-cloud aerosol optical depth (ACAOD) of absorbing aerosols retrieved from the near-UV observations made by the Ozone Monitoring Instrument (OMI) onboard NASA's Aura platform. Physically based on an unambiguous 'color ratio' effect in the near-UV caused by the aerosol absorption above the cloud, the OMACA (OMI Above-Cloud Aerosols) algorithm simultaneously retrieves the optical depths of aerosols and clouds under a prescribed state of the atmosphere. The OMACA algorithm shares many similarities with the two-channel cloud-free OMAERUV algorithm, including the use of AIRS carbon monoxide for the aerosol type identification, CALIOP-based aerosol layer height dataset, and OMI-based surface albedo database. We present the algorithm architecture, inversion procedure, retrieval quality flags, initial validation results, and results from a 12-year long OMI record (2005-2016) including global climatology of the frequency of occurrence, ACAOD, and aerosol-corrected cloud optical depth. A comparative analysis of the coincident and collocated OMACA-retrieved ACAOD and equivalent accurate measurements from the HSRL-2 lidar for the ORACLES phase I operation (August-September 2016) revealed a good agreement (R=0.77, RMSE=0.10). The long-term OMACA record reveals several important regions of the world, including Southeastern Atlantic Ocean, southern Indian Ocean, South-East Asia, tropical Atlantic Ocean off the coast of western Africa, and northern Arabian sea where the carbonaceous aerosols from the seasonal biomass burning and mineral dust originated over the continents are found to overlie low-level cloud decks with moderate (0.3<ACAOD<0.5, away from the sources) to higher levels of ACAOD (>0.8 in the proximity to the sources). No significant long-term trend in the frequency of occurrence of aerosols above the clouds



and ACAOD is noticed when OMI observations that are free from the 'row anomaly' throughout the operation are considered. If not accounted, the effects of aerosol absorption above the clouds introduce low bias in the retrieval of cloud optical depth with a profound impact at increasing ACAOD and cloud brightness. The OMACA aerosol product from OMI presented in this paper offers a crucial missing piece of information of the aerosol loading above cloud that will help us to quantify the radiative effects of clouds when overlaid with aerosols and its resultant impact on cloud properties and climate.

**Keywords:** aerosols above clouds, aerosol absorption, color ratio, Ozone Monitoring Instrument, OMACA



## 1. Introduction

Aerosol-cloud interactions continue to be the most significant source of uncertainty in estimating the role of aerosols and clouds in Earth's changing radiation budget (*IPCC*, 2014). One of the main hurdles is the lack of adequate knowledge of the complex microphysical and optical processes of an aerosol-cloud system that govern the resultant impact on the regional and global climate. An important aspect of the problem is when the light-absorbing aerosols such as biomass burning generated carbonaceous particles and windblown mineral dust overlay low-level cloud decks. Such situations are commonly observed from satellites over several oceanic and continental regions of the world on daily to seasonal scales (*Alfaro-Contreras et al.*, 2015). The transoceanic transport of elevated fine mode aerosols from biomass burning and coarse mode aerosols from dust storms observed from the ground and satellites is a well-known phenomenon (*Prospero et al.*, 2002; *Kaufman et al.* 2005; *Chand et al.*, 2008; *Torres et al.*, 2012). Aerosols in the cloud-free scenario over the dark surfaces are known to produce a net cooling effect (negative radiative forcing) on climate. On the contrary, an overlapping situation of the absorbing aerosols over the cloud can lead to a significant level of atmospheric absorption and thus exert a positive radiative forcing at the top-of-atmosphere (TOA) (*Kiel and Haywood*, 2003; *Chand et al.*; 2009, *Meyer et al.*, 2013; *Feng and Christopher*, 2015). The magnitude of direct radiative effects of aerosols above cloud depends directly on the amount aerosol loading, microphysical-optical properties of the aerosol layer (*Eswaran et al.*, 2015) and the underlying cloud deck (*Meyer et al.*, 2013) as well as geometric cloud fraction and cloud optical depth (*Chand et al.*, 2009). The resultant aerosol-driven atmospheric heating can have a significant influence on the atmospheric stability, cloud formation and lifetime, and the hydrological cycle (*Wilcox*, 2012). Different climate models treat aerosol-cloud interaction processes differently, which results in significant inter-model discrepancies in aerosol forcing assessments, especially over the absorbing aerosol-cloud overlap regions, i.e., Southeastern Atlantic Ocean and South-East Asia (Schulz et al., 2006). To accurately quantify the direct and semi-direct effects of aerosols in the cloudy skies, it is imperative that a measurement-based global database is made available to the community.

Satellite-based passive and active sensors can observe the aerosols in the entire atmospheric column above the cloud deck. Conventional satellite retrievals have provided a global distribution of aerosols over only cloud-free regions, leaving the vast cloudy region unmonitored regarding the presence of aerosols. Likewise, field campaigns in the past mainly focused on the measurements and characterization of aerosol properties in cloud-free conditions to evaluate and improve satellite-based retrievals as well as model simulations. The lack of measurements-based aerosol database in the cloudy atmosphere severely limits our understanding of the aerosol effects on cloud radiative forcing and microphysical properties.

*Torres et al.* (2012) introduced a remote sensing technique to simultaneously retrieve the above-cloud aerosol optical depth (ACAOD) and aerosol-corrected underlying cloud optical depth (COD) using the near-UV observations made by the Ozone Monitoring Instrument (OMI) onboard NASA's Aura satellite. The method is physically established on an unambiguous 'color ratio' effect produced by the absorbing aerosols above clouds in the two near-UV channels, i.e., 354 nm and 388 nm. Under a prescribed state of the atmosphere, the aerosol-led





changes in the cloud radiances measured at the top-of-atmosphere (TOA) relate to a pair of ACAOD and COD. The technique was successfully tested over the case studies of carbonaceous aerosols above the cloud deck found over the Southeastern Atlantic Ocean. Furthermore, ACAOD derived using the near-UV color ratio method was found to be consistent when inter-compared against those inferred from other research-based algorithms applied to the different A-train sensors (MODIS, CALIOP, POLDER) (*Jethva et al.*, 2014).

In this paper, we apply the near-UV color ratio algorithm on a global scale to produce a Level 2 orbital dataset of ACAOD using OMI observations. First, we describe different components of the algorithm (section 2) followed by the theoretical estimates of the expected uncertainties (section 3). Initial validation results of the OMACA aerosol retrieval using airborne HSRL-2 measurements are presented in section 4. The results of the frequency of occurrence analysis of the aerosols above clouds discussed in section 5. A global 12-year long record of ACAOD and aerosol-corrected COD derived from OMACA along with a quantitative analysis of the impact of aerosol absorption on COD retrievals are presented in section 6. The paper is summarized and concluded in section 7.

## 2. Description of the OMACA Algorithm

### 2.1 Physical Basis

Light absorbing aerosols such as carbonaceous aerosols (*Kirchstetter et al.*, 2004) and dust particles (*Wagner et al.*, 2012) exhibit strong wavelength dependence in absorption, particularly in the UV region of the spectrum. On the other hand, clouds show minimal to no spectral dependence in reflectance when measured from the space. When absorbing aerosols overlay the cloud deck, the spectral contrast in the UV is further enhanced producing a strong 'color ratio' effect in the two-channel TOA reflectance measurements (*Torres et al.*, 2012). This effect is often referred to as the 'cloud darkening' caused by the aerosol-led spectral absorption. The UV Aerosol Index (UVAI) measured by OMI is an excellent indicator of the presence of absorbing aerosols in the clear (*Torres et al.*, 1998) as well as the cloudy atmosphere (*Torres et al.*, 2012). Radiative transfer simulation shows that for a prescribed state of the atmosphere, the reduction in the spectral reflectance and UVAI between a pair of wavelengths depends on the optical depth of both aerosols and cloud, single-scattering albedo, and aerosol layer height. Founded on this unambiguous detection of absorbing aerosols above the cloud, *Torres et al.* (2012) introduced a technique that delivers the simultaneous retrieval of ACAOD and aerosol-corrected COD from OMI's two near-UV observations (354 and 388 nm). Figure 1 shows the two-dimensional domain of simulated reflectance at 388 nm (x-axis) and UVAI (y-axis) for several pairs of ACAOD and COD for the carbonaceous aerosols (left panel) and spheroidal dust particles (right panel). Under a prescribed set of assumptions, i.e., aerosol layer height (ALH), aerosol single-scattering albedo (SSA), surface albedo, and geometry, the two-channel measurements of OMI can be related to a pair of ACAOD and COD.





## 2.2 Inputs and Ancillary Datasets

### 2.2.1 Direct Input

We use OMI Level-1b calibrated and geo-located radiance measurements at 354 nm and 388 nm as the primary input to the OMACA algorithm. Foremost, the observed radiances are used to calculate the UVAI (Mie), as

described by *Torres et al.* (2018), which is a residual quantity resulting from the comparison between measured and calculated radiances between 354 nm and 388 nm. Compared to the previously adopted Lambert Equivalent Reflector (LER) based method for calculating UVAI (*Herman et al.*, (1997), *Torres et al.*, (1998)), the new approach offers a better characterization of clouds by accounting for the angular dependence of cloud reflectance (phase function) of liquid water clouds.

### 2.2.2 Ancillary Datasets

#### 2.2.2.1 AIRS CO for Aerosol Type Identification

The aerosol-type identification scheme for OMACA has directly adopted from the operational cloud-free OMI/OMAERUV (version 1.8.9.1) two-channel algorithm. It uses Aura/OMI UVAI coincident with the Aqua/AIRS

retrievals of Carbon Monoxide (CO) to distinguish three major aerosol types, i.e., carbonaceous aerosols, dust particles, and urban/industrial aerosols. OMACA considers two absorbing aerosol types, i.e., carbonaceous aerosols and desert dust. The aerosol type identification scheme adopts a threshold of 0.8 in UVAI to assign either smoke or dust aerosol type that subsequently depends on the columnar amounts of CO retrieved by AIRS. Since the CO is a primary gaseous component of open-field biomass burning, it constitutes a reliable tracer of carbonaceous aerosol.

For the northern hemisphere, the threshold in CO is set to $2.0 \times 10^{18}$ molecules-cm-2, whereas that for the southern hemisphere is $1.6 \times 10^{18}$ molecules-cm-2. For the equatorial region bounded within the latitude range 10°S to 10°N, the aerosol type is determined based on a linearly interpolated CO thresholds.

#### 2.2.2.2 Aerosol Layer Height and Surface Albedo

The representation of aerosol vertical profile relies on an averaged aerosol layer height (ALH) dataset derived using

the 30-month long combined and co-located measurements of CALIOP vertical backscatter and OMI UVAI (*Torres et al*, 2013). The aerosol profile follows a quasi-Gaussian distribution around mean ALH given by the CALIOP-OMI dataset. For the surface characterization, we use near-UV surface albedo database derived using the multiyear OMI LER observations. The method adopts a minimum LER approach which ensures minimal or no contamination from the atmosphere, i.e., aerosols and clouds, in the measured reflectivity. Afterward, the minimum LER dataset

derived from the OMI observations was scaled in the temporal domain to the seasonality of surface albedo retrieved in the visible wavelengths from MODIS. The dataset contains surface albedo values at 354 and 388 nm at a grid resolution of 0.25° x 0.25°. These two components the OMACA algorithm is identical to the ones adopted in the operational cloud-free OMI/OMAERUV two-channel algorithm (*Torres et al.*, 2013).



### 2.2.2.3 *Above-cloud Aerosol Single-scattering Albedo*

The aerosol single-scattering albedo (SSA) is one of the most critical assumptions that can govern the accuracy of ACAOD retrieved from OMI (*Torres et al.*, 2012). A perturbation of +0.03 (- 0.03) in SSA yields about +48% (-25%) error in ACAOD for a reference AOD and COD of 0.5 and 5.0, respectively. The error in ACAOD follows an asymmetric behavior around the perturbed state and is a dynamic function of both aerosol loading (AOD) and underlying cloud brightness (COD). The OMACA algorithm takes advantage of cloud-free standard OMAERUV global retrievals of SSA for characterizing the absorption capacity of aerosols above the cloud. OMI's two-channel OMAERUV algorithm simultaneously retrieves columnar total AOD and SSA at 388 nm in cloud-free conditions on a daily global scale (*Torres et al.*, 2007, 2013). Both retrieved parameters have been evaluated against the ground-based AERONET measurements globally (*Ahn et al.*, 2014; *Jethva et al.*, 2014). *Jethva et al.* (2014) have shown that for carbonaceous and dust aerosol types, which are relevant to the OMACA product, about 52% (77%) of OMI-AERONET matchups agree within the absolute difference of ±0.03 (±0.05). Despite the inherent uncertainties in both inversions, a reasonable agreement between the two independent techniques globally and, in fact, a robust comparison over many important sites affected by biomass burning and dust provided the increased confidence and credibility of the OMAERUV aerosol product.

We have used the existing cloud-free OMAERUV SSA product to generate a daily database to prescribe SSA required in OMACA. The whole world is split up into 14 regions based on the patterns of absorbing aerosols above cloud inferred from the frequency of occurrence analysis (given in section 5). Figure 2 (top) shows the geographical boundaries of the selected regions. For each region and each day of OMI observation, the daily, region-specific value of above-cloud SSA was estimated for the carbonaceous and dust aerosol types separately using the respective cloud-free SSA retrieval weighted by the corresponding UVAI observations (>0.8). Since the OMAERUV algorithm assigns a fixed aerosol type, i.e., smoke, dust, or background aerosol, to each valid cloud-free pixels of OMI, it is possible to estimate daily regional SSA value separately for smoke and dust aerosol types. In the case of missing daily regional SSA due to cloud cover or no OMI orbital data, the method relies on the availability of SSA values on nearby days with a sequential preference, i.e., weekly (±3 days excluding the day in consideration), monthly, or long-term climatology datasets. Observations of aerosols above cloud found outside the boundaries of these 14 pre-selected regions are assigned a fixed SSA of 0.89 and 0.9 for the smoke and dust aerosol types, respectively.

Using ground-based AERONET inversion, *Eck et al.* (2013) showed that the absorption properties of the smoke aerosols over Central/Southern Africa exhibit a seasonal trend where the monthly mean aerosol SSA (440 nm) increases from 0.84 in July to 0.93 in November. An analysis of the OMI-retrieved SSA (388 nm) in the same paper also showed a similar trend as observed by AERONET and suggest that the seasonal change in SSA is widespread over much of southern Africa. The present approach of assigning the above-cloud aerosol SSA captures the daily





variability of aerosol absorption for each region and therefore eliminates the need to assume a time-invariant value of SSA for the retrievals of above-cloud aerosols.

#### 2.2.2.4 *Aerosol Models*

OMACA considers two major partially absorbing aerosol types, i.e., carbonaceous and dust aerosols. The microphysical and optical properties of these aerosol types are adopted from the current operational OMAERUV aerosol models (*Torres et al.*, 2013). Each aerosol type consists of seven distinct sub-models that differ in their imaginary part of the refractive index. The tables included in Appendix list the model properties of both types of aerosol models. Carbonaceous aerosols are assumed as spherical particles with wavelength-dependent imaginary

refractive index in the near-UV region (relative spectral dependence of 20%) that accounts for the presence of organics in the biomass burning generated aerosols (*Kirchstetter et al.*, 2004; *Jethva and Torres* 2011). Dust aerosols are considered spheroidal in shape with a fixed distribution of the spheroidal axis ratio adopted from *Dubovik et al.* (2006) and applied to the near UV observations (*Torres et al.*, 2018).

#### 2.2.2.5 *Look-up-Tables*

OMACA is essentially a look-up-table (LUT) based algorithm. To generate LUT, we employ the vector discrete ordinate radiative transfer model VLIDORT (*Spurr,* 2006). Clouds are assumed to be liquid in phase and follow standard C1 size distribution (*Deirmendjian*, 1969). Aerosol size distribution is assumed to follow a bimodal, log-normal distribution with parameters adopted from the standard OMAERUV aerosol models (see Appendix I).

Carbonaceous aerosols are assumed to be spherical with associated scattering phase functions calculated following the Mie theory. Dust particles, on the other hand, are treated as a mixture of randomly oriented spheroids with a fixed distribution of axis ratios (*Dubovik et al.*, 2006). The phase matrix elements of each spheroidal dust aerosol model of OMACA (Appendix I) were calculated using a set of pre-calculated kernels and associated software package made available by Oleg Dubovik (personal communication). The extracted phase matrix elements of each

dust model were ingested into VLDIORT to simulate TOA radiances. More details on the treatment of spheroidal dust in the OMI aerosol retrieval framework is given in *Torres et al.*, (2018).

LUTs were generated for carbonaceous and dust aerosol models. Each aerosol type consists of seven discrete aerosol SSA (388 nm) ranging from 0.75 to 1.00, for the 354- and 388-nm wavelengths for a total of seven node values in

ACAOD, eight nodes in COD, at different geometries of solar zenith angle, viewing zenith angle, and relative azimuth angle. The simulations were carried out for two surface pressure levels, for four different ALH referenced at respective surface pressure levels, and for five nodes in surface albedo. The node values for each variable are listed in Appendix I. The LUT radiances interpolated at observed geometry, pressure level, ALH, and SSA are matched with the OMI-observed radiance in 2D retrieval domain (Figure 1) to find a pair of ACAOD and COD at 388 nm.

The retrieved values at 388 nm are converted to 354 nm and 500 nm wavelengths following the spectral dependence




of extinction associated with the assumed model in the retrieval process. Figure 3 illustrates a general flow diagram of the OMACA algorithm.

### 2.2.3 Identification of Absorbing Aerosols above Clouds

We adopt a bi-parametric approach to identify the presence of absorbing aerosols above the cloud. The Lambertian

Equivalent Reflectivity or LER measured at a near-UV wavelength is proportional to the brightness of the scene. LER represents the reflectivity of the scene when Rayleigh scattering is taken out from the TOA radiance measurements. On the other hand, UVAI is an excellent indicator of the presence of absorbing aerosols in both cloud-free as well overcast skies (*Torres et al.*, 2012) over all surfaces. Radiative transfer simulations show that while LER is directly proportional to COD, the layers of absorbing aerosols above cloud produces higher

magnitudes of UVAI that depend on the above-cloud AOD, aerosol model, and cloud brightness (COD). Thus, higher values of LER and UVAI potentially represents scenes of absorbing layers of aerosols over low-level cloud deck.

OMI offers a spatial resolution of 13 x 24 km$^2$ at nadir, which intercepts an area of about 338 km$^2$ on the ground for

the VIS part of the instrument (Algorithm Theoretical Document Basis, OMPIXCOR). The ground pixel size and associated area increase significantly at the extreme edge of the swath. A new global product called OMMYDCLD processed in-house co-locates the Aqua/MODIS 1-km cloud retrievals (MYD06) with each OMI pixel footprint (13 x 24 km$^2$) (Joanna Joiner and Brad Fisher, personal communication). OMMYDCLD reports statistics of many MODIS cloud parameters for each OMI footprint such as the median COD, histogram of COD, cloud phase

information, and many others. In addition to this, the OMMYDCLD also provides the total number of MODIS 1-km pixels (clear and cloudy) as well as the total number of cloudy pixels identified at 1-km spatial resolution. The accessibility of these two parameters allows calculating the geometric cloud fraction as observed by MODIS for each OMI pixel. Notice that the current OMACA product does not use the OMMYDCLD product while making above-cloud aerosol retrieval. Instead, we use the information on the geometric cloud fraction derived from

OMMYCLD in the post-retrieval analysis.

### 2.2.4 Algorithm Quality Flags

Each qualified OMI retrieval of the above-cloud aerosols is assigned with an appropriate algorithm quality flag. Table 1 describes the algorithm quality flags attached to each valid retrieval and their associated observed

conditions. Retrievals with the quality flag equal '0' is expected to be the best in quality as they are associated with the larger magnitudes of UVAI (>1.3), and LER388 (>0.25)-both provide high confidence in the detection of absorbing aerosols above the cloud. An analysis using the OMMYCLD product over the southeastern Atlantic Ocean for the period of Jun-July-Aug 2007 revealed a well-constrained non-linear relationship between the MODIS-derived COD times the geometric cloud fraction and LER388. A threshold of LER388 of 0.25 adopted for the

best quality retrievals compares to the COD times the geometric cloud fraction of 3-4. Conversely, given the geometric cloud fraction of unity, the minimum COD retrieved by OMACA would be in the range 3-4.



Lower magnitudes in both parameters might result in lesser confidence in the detection of either overcast pixels (0.20<LER388<0.25, quality flag=1) or the presence of absorbing aerosols above cloud (0.8<UVAI<1.3, quality flag=2). Lower LER values (0.20-0.25) might pose a risk of identifying broken clouds in the OMI pixels, resulting in a geometric cloud fraction lesser than unity-a condition under which the assumption of fully overcast pixel breaks down. Nevertheless, it is also possible that the increased aerosol loading (AOD>2) with a significant absorption capacity (SSA<0.90) above the fully overcast pixels reduces LER measured at TOA (Figure 6 of *Jethva et al.*, 2013). On the other hand, the lower values of UVAI (0.8-1.3) associated with the quality flag '2' may be related to the non-aerosol related artifacts resulting from the inherent uncertainties in the derivation of UVAI. The sources of uncertainties include spectral surface albedo, the unaccounted presence of ice clouds, and viewing geometry of the Sun and satellite. The magnitudes of UVAI depend on several aerosol parameters including ACAOD, COD, SSA, ALH, and spectral dependence of aerosol absorption. Radiative transfer calculations show that for a given value of SSA of 0.90 (388 nm) with an ALH of 3, 4, and 5 km, the UVAI value of 1.3 can be equated to the AOD (388 nm) of 0.30, 0.28, and 0.26, respectively. For a given SSA of 0.84, the values of AOD are 0.22, 0.20, and 0.19. The results of these simulations presented in Table 2 suggest that the minimum value of AOD retrieved using the thresholds in UVAI depends on the actual condition of the scene.

Retrievals assigned with the algorithm quality flag '3' are considered to be the lowest in confidence as it represents spurious non-aerosol related enhancement in UVAI (up to 2.0) at certain extreme geometries. However, if the observed UVAI exceeds a value of 2.0, then the retrievals are assigned with the quality flag '0' or '1' or '2' depending upon the observed LER and UVAI. The OMACA algorithm operates over both ocean and land pixels with terrain pressure greater than 800 hPa, which encompasses the majority of the regions of frequent aerosols-cloud overlap (see section 5). Retrievals over oceanic cloud pixels are performed at all Sun glint angles if measured LER exceeds 0.30; for the 0.20<LER<0.30 condition retrievals are performed with pixels having Sun glint angle>20° to avoid glint-related artifacts in the retrievals.

## 3. Uncertainty Estimates

The OMACA algorithm relies on the presumptions about the atmosphere and surface properties. It is, therefore, imperative to estimates the sensitive of the OMACA retrievals to the departure of the actual state of the atmosphere from the one assumed in the inversion. Earlier, Torres et al., (2012) described the theoretical uncertainties in the near-UV based retrieval of ACAOD and aerosol-corrected COD. The analysis was, however, confined to a restrictive range of input conditions. Here, we re-perform the same exercise by considering an extended range of perturbation in each assumed parameter.

The approach to calculating the uncertainties in the OMACA retrievals follows a perturbation-based method. The errors were calculated by contrasting the retrievals derived assuming a reference state and perturbed state of a particular input parameter. For example, given a fixed set of aerosol size distribution, ALH, and surface albedo,



OMACA retrievals are derived assuming a range of aerosol SSA. The retrievals then compared with those derived considering a reference value of SSA, which is the center value of the prescribed range. The errors in the retrievals then can be calculated given both underestimated and overestimated values of an assumed parameter.

Table 3 lists the percent error in ACAOD (388 nm) caused by varying uncertainty in the aerosol SSA for an above-cloud smoke situation. The reference value of SSA (388 nm) was assumed to be 0.89; ALH and surface albedo were referenced at 3.0 km and 0.05, respectively. Errors were calculated for a range of uncertain SSA, i.e., -0.05 (underestimation) to 0.05 (overestimation) in step of 0.01. The optical depth of cloud underneath the aerosol layer was assumed to be 10. The errors in ACAOD are found to behave non-linearly to the perturbations in the assumed SSA. It also depends on the true value of ACAOD. Furthermore, the error conforms to an asymmetric behavior around the reference value of SSA; larger magnitudes of error are associated with the overestimated SSAs, whereas relatively lower errors are obtained when SSA was underestimated. The selection of above-cloud SSA values in actual OMACA retrievals relies on a daily, regional database of cloud-free SSA values retrieved from the standard OMAERUV aerosol product, as described in section 2.2.2.3. The accurateness of assigned above-cloud SSA, therefore, depends on the accuracy of cloud-free OMAERUV SSA retrievals as well as on the validity of the assumption that aerosol absorption properties are invariant between cloud-free and above-cloud aerosols scenes.

An intercomparison analysis of OMI-AERONET SSA retrievals from over 269 AERONET sites distributed globally showed an agreement within ±0.03 and ±0.05 limits for about 51% and 76% of total 5463 collocated matchups (*Jethva et al*., 2014). When segregated by the range of AOD (440 nm) and UVAI, 49% (AOD<0.7, UVAI<1.0) and 53% (AOD>0.7, UVAI>1.0) of the total OMAERUV-AERONET SSA (440 nm) retrievals are found to agree within their estimated uncertainties of ±0.03. The agreement improved to 74% and 79%, respectively, when the uncertainty limit was relaxed up to ±0.05. The statistical comparison was found even better when the matchups were segregated by the aerosol type, i.e., only smoke or dust, over many long-term sites located in the biomass burning and dust dominated regions.

The AERONET-OMI SSA comparative analysis cannot be treated as an actual "validation" of the satellite retrievals as both datasets are essentially derived from the two different platforms using fundamentally different algorithms that rely on a set of assumptions. However, a robust agreement between the two SSA datasets at larger AOD (>0.7) and UVAI (>1.0) established consistency and provided an increased level of confidence in the OMAERUV absorption retrievals in cloud-free areas. Therefore, we expect that the above-cloud SSA values assigned in the OMACA algorithm over different regions should be accurate within ±0.03. Table 3 shows that uncertainty in the assumed SSA of +0.03 (-0.03) leads to an error in the retrieved ACAOD by +42% to +46% (-20% to -25%). The estimated errors are much larger (90%-100%) given the larger uncertainty (±0.05) in the assumed SSA.

Relative to the errors in ACAOD due to the uncertain SSA, departures of ALH from the assumed state results in lower errors (Table 4). For assigning the ALH, OMACA relies on a global, monthly mean dataset derived from the





30-month coincidence and collocated CALIOP and OMI observations (*Torres et al.*, 2013). The expected uncertainty in the derived ALH dataset is about ±1 km for which the error in ACAOD could vary between -5% to -13% and +8% to +21% for an overestimated and underestimated ALH by +1 km and -1 km, respectively.

The OMACA 2D retrieval domain shown in Figure 1 suggests that the retrieved value of ACAOD is primarily modulated by the magnitudes of UVAI. Several parameters including ACAOD, COD, ALH, SSA, and AAE can influence the magnitudes of UVAI. For instance, given a fixed set of spectral AODs at 354 and 388 nm wavelengths, ALH, and SSA at 388 nm, the magnitude of derived UVAI strongly varies with the assumed value of AAE (Figure 4 of *Jethva and Torres*, 2011). In other words, for a given value of observed UVAI, different

assumptions of AAE would result in different values of the retrieved AOD. The "smoke" and "dust" aerosol models adopted in the OMACA algorithm assume an invariant spectral dependence of the imaginary part of the refractive index. For the carbonaceous and dust aerosol models, the relative spectral dependence in the imaginary index is assumed to be +20% (*Kirchstetter et al.*, 2004; *Jethva and Torres*, 2011) and 39%, respectively, between the 354 and 388 nm wavelengths. For a fixed set of size distribution parameters and range of SSA (388 nm), this results in

AAE in the range 2.5-3.0 for the carbonaceous aerosol models, and 2.0-4.0 for the dust models (see Appendix I). Theoretical error (%) in ACAOD (388 nm) due to the uncertainty in the assumption of aerosol AAE (354-388 nm range) is listed in Table 5. The reference value of AAE was assumed to be 2.65 corresponding to the moderately absorbing smoke model, and COD underneath the aerosol layer was assumed to be 10. AAE was perturbed in steps of 0.5 in both directions from the reference value. Similar to the uncertain SSA simulations, errors in ACAOD

behaves asymmetrically to the perturbations in AAE with larger (relatively lower) magnitudes of errors are associated with the underestimated (overestimated) AAE.

The sensitivity of ACAOD retrieval to the uncertainty in three major assumptions made in OMACA considered a broad range of perturbation. However, we anticipate that the prescribed values of SSA, ALH, and AAE are accurate

to within ±0.03, ±1 km, and ±0.5, respectively, for which the errors in ACAOD can vary from -23% to +46% in the ACAOD range 0.25-1.0. In the situation when the uncertainty in the assumed inputs leads to errors of opposite sign, the resulting error in the retrievals is likely to be lower than expected due to the cancellation of individual errors. On the other hand, an agglomeration of the errors of the same sign can further amplify the overall uncertainty in the retrievals. Nevertheless, it is practically hard to arrive at the actual uncertainty in the OMACA retrieval for every

pixel due to the ill-posed nature of the inversion problem. Evaluating the accuracy of the satellite retrievals requires an independent set of direct measurements of aerosols, in this case above the cloud, discussed in the following section.

## 4.    Preliminary Validation

Unlike the validation exercise of satellite-based aerosol retrievals in cloud-free skies for which the ground-based direct measurements of AOD are amply available from hundreds of sites worldwide such, an assessment of ACAOD retrieved from the satellite is a challenging task due to the lack of such reference aerosol measurements above the

clouds. This is because field campaigns in the past mainly focused on the measurements and characterization of aerosol properties in cloud-free conditions leaving vast cloudy areas unmonitored regarding the aerosol measurements. However, the airborne lidar such as High Spectral Resolution Lidar (HSRL) when flying above the top of the aerosol layer can make direct measurements of aerosol extinction and thus provides AOD above the cloud.

Also, airborne sunphotometers can make such measurements by flying above the cloud and below the aerosol layer. Using a limited data set of the direct measurements of AOD above the cloud carried out by the NASA Ames Airborne Tracking Sunphotometer (AATS) and Sky-Scanning, Sun-Tracking Atmospheric Research (4STAR) sensors during different field campaigns, *Jethva et al.* (2016) have validated ACAOD retrieved using the 'color ratio' method (*Jethva et al.*, 2013), similar to the one presented here, but applied to the TOA visible-near Infrared

reflectances measured by the MODIS.

NASA's ORACLES-ObseRvations of Aerosols above CLouds and their intEractionS (https://espo.nasa.gov/oracles) is an ongoing multi-year field experiment supported by the NASA Earth-Venture Suborbital Program. ORACLES intended to make accurate airborne remote sensing and in situ measurements of aerosols and clouds in the

Southeastern Atlantic Ocean. At the time writing this paper, ORACLES has already completed the two phases of its operation, 1$^{st}$ phase conducted in August-September 2016 and the 2$^{nd}$ phase in August 2017. During the 1$^{st}$ phase, HSRL-2 lidar developed by NASA Langley Research Center made extensive measurements of smoke aerosols, including detailed vertical measurements of aerosol backscatter, extinction, and AOD at 355 and 532 nm, above shallow marine clouds while deployed from NASA ER-2 aircraft. HSRL-2 measurements of AOD are routinely

compared with that of AERONET and found to agree well (R=0.98) with the latter. Taking advantage of highly accurate and valuable dataset of AOD above the cloud, we evaluate the ACAOD retrievals from OMACA for the cases of coincident and collocated OMI-HSRL2 measurements.

Figure 4 shows the comparison of spectral ACAODs measured by HSRL-2 and retrieved from OMI/OMACA for a

total of seven ER-2 flights conducted during August-September of 2016 (August 26, September 12, 16, 18, 20, 22, 24). HSRL-2 measurements falling within the boundaries of each OMI pixel, as defined in the OMPIXCOR product, were averaged and compared with the ACAOD value of the corresponding OMI pixel.  The spatially collocated aircraft-satellite matchups were grouped according to the three different time windows, i.e., no time constraints (in hours), and ΔT of ±2 hours and ±1 hours, where ΔT is the absolute time difference (in hours) between the OMI

overpass and HSRL-2 measurements. To facilitate the direct comparison, ACAODs from HSRL-2 were interpolated to the OMI wavelengths of 388 nm and 500 nm following the Angstrom Exponent calculated using the 355-532 nm measurements. For the 'no time constraint' group, the collocation procedure yields more than 500 matchup data points with a correlation, RMSE, mean bias of 0.676, 0.23, and -0.11, respectively, at 388 nm. The comparison, however, significantly improves when the matchups are restricted to the narrower time windows of ΔT of ±2 and ±1

hour. For the ΔT=±1-hour matchup group, the comparison yields correlation, RMSE, and mean bias of 0.77, 0.1, and 0.007 with the slope and intercept of 0.6 and 0.19 of the linear regression. Aerosol mass is in constant motion depending on wind speed and direction and therefore, allowing a wider time window between the satellite and





aircraft measurements may end up in a mismatch which appears to the case for relatively poorer comparison when no time limits were imposed on the comparison. Noticeably, the OMI-HSRL2 comparison of ACAOD provides the best agreement at the 500-nm wavelength, where the OMACA does not perform inversion but reports ACAOD based on the spectral dependence of extinction assumed in the aerosol model.

The remaining discrepancies in the comparison could be primarily attributed to the inherent uncertainties associated with both types of measurements, particularly in the satellite retrievals of ACAOD as discussed in the previous section, and spatiotemporal heterogeneity in aerosol fields unresolved by the collocation method. Despite these uncertainties, a reasonable agreement of OMACA-retrieved ACAOD with more accurate measurements from

HSRL-2 for the ORACLES campaign provided the credibility and confidence in the product and allowed us to use it for the regional and global analyses presented in the rest of the paper.

## 5. The frequency of Occurrence of Absorbing Aerosols above Clouds

### 5.1 Spatial Distribution

The regional and global climate impact of absorbing aerosols above cloud depends on the total aerosol loading

above the cloud, the microphysical and optical properties of aerosols and underlying cloud deck as well as the spatial and temporal extent of the aerosol-cloud overlap scenes. We have carried out a global frequency of occurrence of absorbing aerosols above the cloud (FOACA) to identify the regions of frequent aerosols-cloud overlap. We adopt a bi-parametric approach to identify the scenes of absorbing aerosols overlaying the low-level cloud decks, as described in section 2.2.3. FOACA is referenced to the cloudy-sky observations and defined as the

ratio of the total number of days with an ACA condition (LER>0.25 & UVAI>0.8) to the total number of days with cloudy-condition (LER>0.25). Additionally, we take advantage of the OMMYDCLD product to calculate the geometric cloud fraction for each qualifying OMI pixel. For the FOACA analysis, we adopt a less strict threshold of cloud fraction of 0.5. Since the main purpose of this analysis was to identify the presence of absorbing aerosols and cloud in the same atmospheric column and not to quantitatively retrieve ACAOD, a less strict value cloud fraction

should adequately represent both aerosols and clouds in the corresponding pixel.

Figure 5 shows the monthly averages of FOACA derived from the 12-year record (2005-2016) of OMI following the above-described method. This analysis reveals several important regions of the world where the overlap of absorbing aerosols over clouds are frequently observed. During July through September, carbonaceous aerosols

generated from agricultural burning over the central/southern Africa are mobilized over the semi-permanent low-level stratocumulus water clouds in the southeastern Atlantic Ocean (*Torres et al.*, 2012; *Alfaro-Contreras et al.*, 2015; *Meyer et al.*, 2015). With more than 80% of the cloudy-sky observations are identified as aerosols above cloud during the northern hemisphere summer, the southeastern Atlantic Ocean is considered to be one of the prime regions the world and also a natural laboratory to study the aerosols above cloud phenomenon. The springtime

biomass burning activities such as burning of forest, savanna/grassland, and crop residue over Southeast Asia (SEA) countries, including Thailand, Myanmar, Laos, Cambodia, and Vietnam release significant amount of trace





gases and carbonaceous aerosols in the atmosphere (*Elvidge and Baugh*, 1996; *Streets et al.*, 2003). Natural color images from satellite show that smoke particles emitted from these activities were mobilized under the influence of winds over the widespread cloud deck over southern China, creating the appearance of the cloud deck darker. FOACA results show that about 20%-40% of the cloudy days are marked with smoke aerosols overlying bright

cloud deck. Also, the smoke-cloud overlap seen in the OMI data is not merely confined to over land, but also extended over the western Pacific Ocean albeit less often.

Dust storms originated over the Sahara Desert in North Africa during the summer are often transported across the tropical Atlantic Ocean (*Prospero et al.*, 2002; *Kaufman et al.*, 2005; *Huang et al.*, 2010; *Yu et al.*, 2015). A

substantial part of the dust transport occurs over the low-level stratocumulus clouds. The FOACA analysis shows that the presence of mineral dust aerosols above the cloud accounts for 40%-80% of the total cloudy-sky OMI observations. Similarly, mineral dust particles originated from the Arabian Peninsula, Iran and Pakistan often mobilized over the Arabian Sea during the active months (July-Aug) of southwest Indian summer monsoon (*Moorthy et al.*, 2005; *Satheesh et al.*, 2006). The transported dust frequently overlies the low-level clouds such that

the dust-cloud overlap accounts for the 40%-60% of the cloud-sky days.

The springtime Asian outflow of air pollutants, both anthropogenic and mineral dust, across the Pacific Ocean is documented in various studies (*Liu et al.*, 2003; *Huebert et al.*, 2003). The FOACA maps for the April and May months reveals that the transport pattern encompasses the entire northern Pacific basin from the eastern coast of Asia

to the western coast of North America with 10%-30% of the cloudy-sky scenes identified as aerosols over the clouds. One of the salient features of the FOACA analysis is the smoke transport from southern Africa to the Indian Ocean. Carbonaceous aerosols emitted from the fires in southeastern Africa during the late biomass burning season (September), are often taken away from the continent along the eastward flow and advected above the clouds over the southern Indian Ocean. The FOACA map for the month of September highlights that the transport of smoke

over the cloud is confined to within the latitude range 35°S to 20°S with 40%-50% (20%-30%) cloudy scenes marked by the smoke overlaying clouds just off the coast of southeastern Africa (over the western coast of Australia).

### 5.2    Long-term Trends in FOACA

Since mid-2007, OMI observations have been affected by a possible external obstruction that perturbs both the measured solar flux and Earth radiance. This obstruction affecting the quality of radiance at all wavelengths for a particular viewing direction is referred to as "row anomaly" since the viewing geometry is associated with the row numbers on the charge-coupled device detectors. The row anomaly issue was detected first time in mid-2007 for few rows which over the period of operation expanded to other rows in 2008 and later. Figure 6 (a) (top panel) shows the

current status of the row anomaly as identified by the anomaly algorithm developed for the NASA OMI total ozone product OMTO3 (*Schenkeveld et al.*, 2017). At present, about half of the total 60 rows across the track are identified and flagged as row anomaly affected positions for which no physical retrievals are being performed.





Earlier, it was learned and concluded that for the derivation of a meaningful trend in the global FOACA only those rows or positions of the OMI instrument should be considered that are mostly unaffected by the row anomaly throughout the OMI operation period (*Jethva, H.*, 2015). This is because UVAI exhibits a dependency on the scan

angle in which the westward viewing geometry (left side of the scan, row numbers 1-30) shows higher values than those measured for the eastward-looking geometry (right side of the scan, row number 31-60). The row-averaged UVAI for the OMI operation period shown in the middle panel of Figure 6 (b) reflects the asymmetry in UVAI. OMI lost its most rows on the right side of the scan post-2007 operation due to the row anomaly. The remaining rows on the left side of the scan, where the UVAI values are larger than those on the right side of the scan,

introduces a positive shift in the absolute values of UVAI which leads to an overall positive trend in the FOACA. The global trend (%) in FOACA calculated using all rows of OMI, therefore, gives a positive trend (0.178 per year) as shown in the bottom panel of Figure 6 (c). However, when the global trend is calculated using observations from rows which are mostly free of row anomaly throughout the OMI operation period, i.e., row number 1-23, the FOACA shows a negligible trend (0.022 per year). Regionally, we find similar results of positive trends in FOACA

when all rows of OMI were used in the calculations (not shown here) against the statistically negligible trend when observations from rows 1-23 were considered, as shown in Figure 7. Based on the present findings and also according to the results published in an interactive comment (*Jethva, H.*, 2015), we strongly recommend the users of OMACA product to use only those observations that are free of row anomaly throughout the OMI operation (2004 to present) for the trend related analysis.

## 6. Results from 12-year Long OMACA Record

### 6.1 Global Distribution of Above-cloud AOD

Figure 8 shows the global distribution of ACAOD (388 nm) derived from OMACA algorithm applied to the 12-year long record of OMI observations. The OMACA Level 2 retrievals with algorithm quality flag '0', '1', and '2' were averaged on every 0.5° x 0.5° spatial grid to derive global monthly gridded dataset. Also, a threshold of 0.75 in the

geometric cloud fraction calculated using the OMMYDCLD product was used to filter out the L2 pixels with broken cloud fields. Only those grids are considered in the long-term monthly averages for which 1) the total number of days with valid retrievals for a particular month is greater than 3, and 2) number of years greater than 3 out of the 12-year record. The distribution plot reveals moderate to high aerosol loading above the clouds over several regions of the world.

During the northern hemisphere summer, larger magnitudes of ACAOD (>0.7) are retrieved over the Southeastern Atlantic Ocean along the coast of Namibia and Angola. Noticeably, the area coverage of ACAOD expands substantially as the season progresses with retrieval coverage confined to within 1500 km from the western coast of Africa in June to encompassing almost the entire Atlantic Ocean basin (~5000 km) in September. Largest

magnitudes of ACAOD are observed in August and September when the biomass burning activities also peak in the central/southern Africa. In March and April, biomass burning in Southeast Asia emit large amounts of carbonaceous





aerosols, which under the influence of winds are transported above the bright cloud deck over southern China, where OMACA retrieves large values of ACAOD (>0.8) in the vicinity of burning areas. Noticeably the area-coverage of aerosol-cloud overlap extends far from the source burning region to over the East China Sea albeit with a decreasing gradient in the retrieved ACAOD. During the late biomass burning season (September), carbonaceous aerosols

emitted from the fires in southeastern Africa often drift away from the continent along the eastward flow (*Garstang et al.*, 1996), and advected above the clouds over the southern Indian Ocean. The spatial pattern of retrieved ACAOD encompasses the entire Indian ocean basin stretching from the southeastern coast of Africa with ACAOD in the range 0.4-0.5 to the western coast of Australia with reduced ACAOD in the range 0.2-0.3.

During the northern hemisphere summer months, ACAOD in the range 0.3-0.5 is observed over the tropical Atlantic Ocean where the transport of dust takes place from the Saharan desert to over the oceanic clouds. The area coverage of the retrievals is maximum in July spanning half of the tropical Atlantic Ocean basin with maximum ACAOD (~0.5) just off the coast of northern Africa. Also, a gradient in ACAOD is noted over the northern Arabian Sea during summer, owing to the dust transport from the Arabian region to over the low-level clouds over the ocean.

OMACA product also captures springtime (April and May), long-range trans-Pacific transport of dust aerosols originated over the Gobi and Taklamakan Deserts possibly mixed with urban pollution and smoke along the transport pathways and over the clouds. The magnitude of retrieved ACAOD of the above-cloud Asian outflow ranges from 0.4-0.5 near the eastern coast of North-East Asia, reducing to 0.2-0.3 along the transport over the mid-

Pacific and up to the western coast of North America.

### 6.2    Regional Time-series of Above-cloud AOD

Figure 9 (left y-axis, color: red) shows the regional, monthly mean time-series of ACAOD (388 nm) for the five prominent regions of the world where the frequent overlap of absorbing aerosols above the cloud is observed. The

regional monthly mean ACAODs were calculated in the following way. For each region and month, an averaged value of ACAOD was calculated, and a number of Level 2 observations that went into the averaging was also stored. Subsequently, a set of 12 monthly-averaged values scaled by multiplying them with a fraction which is defined as the ratio Level 2 observations for the individual months to the maximum number of Level 2 observations found over the 12-year period (2005-2016) over the same region. Scaling the monthly-averaged value with the calculated

fraction ensures the representativeness  of ACAOD statistics over the spatial and temporal domains and therefore facilitates the intercomparison. For instance, a comparison of the two averaged values derived from two different set of statistics, i.e., under-populated and adequately-populated, likely results in the misinterpretation of monthly time-series data. The procedure described above was applied to the OMACA observations for the five prominent aerosol-cloud overlap regions to produce time- series shown in Figure 9.

The temporal evolution of ACAOD (388 nm) over the Southeastern Atlantic Ocean exhibits a repetitive seasonal cycle with monthly mean values reaching up to 0.4 during the dry biomass burning season. Although an inter-annual





variation is apparent in the time-series, e.g., lower and higher ACAOD during the burning season of 2012 and 2015, no significant trend is noticed over the 2005-2016 OMI record. Over the tropical Atlantic Ocean, the monthly ACAOD values fall in the range 0.2-0.3 during summer months when dust aerosols from northern Africa are mobilized over the low-level oceanic stratocumulus clouds. The inter-annual variations of ACAOD over the Arabian

Sea are found to be significant with the monthly value of 0.4 during the first two years of the record (July and Aug of 2005 and 2006) followed by a drastic reduction in the aerosol loading above the cloud during the middle part of

the record. The springtime biomass burning and resulting smoke aerosols above cloud over South-East Asia (4[th] panel) is evident in the time-series where the peak values of monthly ACAOD vary from 0.2 to 0.5 depending on the year.

It is important to quantify the fraction of the total columnar aerosol loading situated above the cloud for several applications. For instance, the cloud-free aerosol retrieval represents the aerosol in the entire vertical column while avoiding the cloudy-sky scenes. Therefore, the statistics of cloud-free retrievals over the regions with frequent aerosol-cloud overlap becomes restrictive, leading to the partially incomplete representation of aerosol properties on

a regional and temporal scales. This kind of scenario affects the calculations of regional climatology, radiative forcing assessments, and aerosol transport. However, the availability of above-cloud (OMACA) and cloud-free total columnar (OMAERUV) AODs from OMI allows us to estimate the fractional aerosol loading above the cloud. The ratio of monthly mean ACAOD to the total columnar AOD (both at 388 nm) displayed on the right-side y-axis of Figure 9 shows that the fractional aerosol loading over the cloud can be as large as 80%-100% during the peak

months of biomass burning and dust transport over the respective regions. A significant fraction of the aerosol column above clouds indicates that the long-range transport of partially absorbing aerosols occurs in the free troposphere and over the low-level clouds. Although the above-cloud and total column AOD are comparable in magnitude during peak aerosol activities,

### 6.3   Regional Time-series of Aerosol-corrected Cloud Optical Depth

Figure 10 displays the monthly mean evolution of aerosol-corrected COD (left y-axis, color: red) for the five prominent regions of aerosol-cloud overlap. The monthly mean values were calculated following the procedure described in the previous section. The seasonal cycle of COD over the Southeastern Atlantic Ocean exhibits a repetitive behavior with a maximum value of COD reaching up to 8.0 during the peak burning period. On the other hand, the monthly cycle of COD over the tropical Atlantic Ocean (Arabian Sea) during the same season shows more

considerable interannual variations with COD in the range 6 to 14 (4 to 10). The magnitudes of aerosol-corrected COD over South-East Asia during the springtime biomass burning season (March-April) are found to be the largest among five regions considered here with values ranging between 16-20 except for the years 2008 and 2013 when COD was lesser than 12.

The right-side y-axis of Figure 10 (color: blue) depicts the concurrent monthly time-series of the percent difference between apparent (non-corrected) and aerosol-corrected CODs the magnitudes of which represent the error in the



retrieval of COD at 388-nm wavelength with reference to the corrected COD if the presence of aerosols is ignored in the inversion. Note that only those COD retrievals are considered here for which the absorbing aerosols are identified above the clouds. While there is a significant variation in the magnitudes of the % difference between the two CODs over different regions, generally larger errors are associated with the higher aerosol-corrected COD and

ACAODs (shown in Figure 9). More discussion on the impact of aerosol absorption on cloud retrievals is presented in the next section.

### 6.4   Impact of Aerosol Absorption on Cloud Retrievals

The presence of absorbing aerosols above cloud obstructs the light reflected by the cloud top, and thus reduces cloud-reflected upwelling UV (*Torres et al.*, 2012), VIS, and NIR radiation (*Jethva et al.*, 2013; *Meyer et al.*, 2015)

reaching the TOA. Therefore, cloud retrievals of COD derived from passive sensors such as OMI are expected to be biased low if absorbing aerosols are not accounted for in the inversion. OMACA product reports two sets of COD, one corrected for the presence of absorbing aerosols overlying cloud deck, and one retrieved assuming no aerosols above the cloud which is termed as the apparent COD. The magnitudes of bias in the apparent COD depend on the strength of aerosol absorption and backscattering as well as on the actual value of COD. Note that OMACA does not

directly retrieve the aerosol absorption optical depth (AAOD) but retrieves ACAOD assuming an a priori value of SSA (see section 2.2.2.3). Therefore, the AAOD can be readily calculated using these two pieces of information as AAOD = ACAOD*(1- SSA). Figure 11 shows the percent bias in COD (388 nm), defined as (Apparent COD – Aerosol- corrected COD)/Aerosol-corrected COD *100, as a function of concurrent AAOD (388 nm). The percent bias was calculated for the distinct range of aerosol-corrected COD and the two aerosol types, i.e., smoke (left

panel) and dust (right panel). All OMACA Level 2 orbital data (2005-2016) for the respective regions and the two aerosol types were accumulated separately and subsequently averaged as a function of corresponding AAOD bins of a sampling size of 5000 retrievals.

For both aerosol types, increasing the magnitude of negative bias in the retrieval of apparent COD is related to

AAOD suggesting the impact of aerosol absorption on the retrieved COD when the presence of absorbing aerosols is ignored in the inversion. Retrievals identified with the 'smoke' aerosol type, predominantly found over the biomass burning regions of Southeastern Atlantic Ocean and South-East Asia show a much larger range of AAOD and associated bias in COD than that observed with 'dust' aerosols found in the regions of dust transport over the tropical Atlantic Ocean and the Arabian Sea. Noticeably, the magnitudes of negative bias in the cloud retrievals are

also co-dependent on the absolute values of COD (here it is aerosol-corrected COD). For instance, for an AAOD of 0.1, the bias in COD is ~ -25% at a lower range of COD (5-10), which gets twofold in magnitude (~ -50%) at the higher range of COD (20-50).

### 7.   Summary and Concluding Remarks

We have developed a global above-cloud aerosol algorithm, formally named as OMACA (OMI Above-cloud

Aerosols), to simultaneously derive the columnar optical depth of absorbing aerosols above the cloud and underlying aerosol-corrected cloud optical depth from the near-UV observations made by Aura/OMI. Physically




based on the enhanced spectral contrast in the near- UV region (354-388 nm) caused by aerosol absorption above the cloud, OMACA relates the TOA observations in the two channels to a pair of ACAOD and COD under a prescribed set of assumptions. The architect of the OMACA algorithm in terms of the ancillary datasets (CALIOP-OMI based ALH, OMI-based near-UV surface albedo, and use of AIRS CO for the aerosol type identification),

aerosol models (smoke and dust), and retrieval approach (two-channel inversion) is analogous to the OMI's two-channel, cloud-free OMAERUV aerosol algorithm. OMACA was applied to the entire record of OMI observations (Oct 2004 to present) to deduce a global research product of AOD above the cloud. Currently, the Level 2 orbital data product is stored on a freely accessible Aura Validation Data Center webpage (https://avdc.gsfc.nasa.gov/pub/data/satellite/Aura/OMI/V03/L2/OMACA/). Also, the OMACA product is produced

in the forward processing mode with a maximum latency of about three days, which is associated with the availability of AIRS L3 CO data for the aerosol type identification.

An analysis of the frequency of occurrence of the above-cloud absorbing aerosols reveals several important regions of the world where the overlap of absorbing aerosols and cloud are frequently observed on a monthly to seasonal

scales. These regions include Southeastern Atlantic Ocean and Southeast Asia where layers of smoke aerosols produced from the seasonal agricultural biomass burning spread thousands of kilometers over the regional low-level stratocumulus cloud deck; the tropical Atlantic Ocean and the Arabian Sea where dust aerosols transported from Sahara and Arabian deserts, respectively, found over low-level clouds; the northern Pacific Ocean where dust particles originated from Asian deserts possibly mixed with the pollution haze along the eastward transport

pathways are found to overlie clouds; and the southern Indian Ocean where the smoke produced from agricultural burning over southeastern Africa drifts along the easterly winds and overlie oceanic cloud deck. Globally as well as on a regional scale, no significant trend in the frequency of ACA was noted when only those observations of OMI instrument (Row # 1 to 23) that are free of row anomaly throughout the OMI operation period (2004 to present) were used in the calculation. The climatology maps of the retrieved ACAOD (388 nm) derived from a 12-year long

OMACA record show moderate (0.3<ACAOD<0.5, away from the sources) to higher aerosol loading (ACAOD >0.8 in the proximity to the sources) above the cloud over these prominent aerosols-cloud overlap regions. When compared with the cloud-free, columnar aerosol loading distribution retrieved from the OMI/OMAERUV two-channel algorithm, ACAOD accounts for as large as 60%-100% of the total columnar loading over different regions during peak biomass burning and dust transport seasons.

The aerosol-corrected CODs retrieved at the near-UV wavelength (388 nm) are found to be noticeably higher than those retrieved assuming no aerosols above the cloud. The percent bias in COD with reference to the aerosol-corrected COD is found to strongly correlate with AAOD as well as the magnitude of COD. For instance, carbonaceous aerosols above cloud found over the Southeastern Atlantic Ocean and South-East Asia during

respective biomass burning seasons result in a significant negative bias in apparent COD the magnitudes of which increase with increasing aerosol absorption as well as the cloud brightness.





A direct comparison of coincident and collocated ACAODs derived from OMI/OMACA and that measured from airborne HSRL-2 measurements for the ORACLES phase I operation (August-September 2016) over the Southeastern Atlantic Ocean showed a good level of agreement with a correlation and RMSE of 0.7 and 0.1 respectively. We further plan to extend the validation of OMACA using the direct measurements of ACAOD from

airborne 4STAR sunphotometer operated from NASA's P3-B aircraft during ORACLES phase I and II. Additionally, a suite of aerosol microphysical and optical measurements from the ORACLES campaign, particularly those characterizing spectral aerosol absorption, will help to verify and improve the region-specific aerosol models employed in OMACA.

Conventional aerosol remote sensing algorithms provide distribution of aerosols in the cloud-free areas leaving behind vast cloudy regions unmonitored regarding the co-presence of aerosols and clouds. OMACA aerosol product offers a quantitative characterization of aerosol loading above cloud on a global scale. Several observational and modeling studies have shown that an overlap of absorbing aerosols above cloud leads to significant atmospheric warming, which can affect cloud lifetime and hydrological cycle. The magnitudes of these effects depend upon the

amount of aerosol loading above the cloud, cloud brightness (COD), optical and microphysical properties of aerosols and clouds, and cloud fraction. The OMACA aerosol product from OMI presented in this paper offers a crucial missing piece of information of the aerosol loading above cloud that will help us to quantify the radiative effects of aerosols above the cloud and its resultant impact on clouds and thus climate. A global above-cloud aerosol product in conjunction with the standard cloud-free aerosol product provides us with an unprecedented all-sky

aerosol distribution from space. This can substantially enhance our knowledge of how aerosols affect cloud radiative forcing and microphysical properties, and aerosol transport.





**Acknowledgments**

Authors acknowledge the support of OMI SIPS team (https://earthdata.nasa.gov/about/sips/sips- omi) for processing the OMACA dataset. We thank Dr. P. K. Bhartia at NASA Goddard Space Flight Center for his guidance and helpful suggestions throughout the development of the OMACA product. Peter Leonard at ADNET Systems Inc has

5    extended help in coding the software package. Dr. Oleg Dubovik at the University of Lille1 provided a standalone software package for the simulation of phase matrix elements of spheroidal particle size distribution used to generate the dust aerosol look-up-table. Acknowledgements are due to the HSLR team members for conducting the measurements of aerosols above cloud during ORACLES phase I operation, which helped validate the OMACA product over the Southeastern Atlantic Ocean.



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





## Appendix I

### Optical and microphysical properties of OMACA Dust Aerosol Model

| Model# | Imaginary Index | | | Single-scattering Albedo | | | Absorption Angstrom Exponent | |
|--------|------|------|------|------|------|------|------|------|
| | Wavelength in nm | | | Wavelength in nm | | | | |
| | *354* | *388* | *500* | *354* | *388* | *500* | *354-388* | *354-500* |
| 1 | 0.02303 | 0.01662 | 0.00720 | 0.74982 | 0.77921 | 0.86268 | 1.97901 | 2.34458 |
| 2 | 0.01279 | 0.00923 | 0.00400 | 0.80740 | 0.83778 | 0.91046 | 2.49312 | 2.82591 |
| 3 | 0.00832 | 0.00600 | 0.00260 | 0.84727 | 0.87606 | 0.93640 | 2.90115 | 3.14519 |
| 4 | 0.00561 | 0.00405 | 0.00176 | 0.88062 | 0.90532 | 0.95430 | 3.15286 | 3.38913 |
| 5 | 0.00256 | 0.00185 | 0.00080 | 0.93213 | 0.94886 | 0.97805 | 3.71332 | 3.87787 |
| 6 | 0.00128 | 0.00092 | 0.00040 | 0.96221 | 0.97234 | 0.98620 | 4.03071 | 3.52634 |
| 7 | 0.00000 | 0.00000 | 0.00000 | 1.00000 | 1.00000 | 1.00000 | N/A | N/A |

Real refractive index = 1.55
Mean Radius (fine mode) = 0.052 μm; Mean Radius (coarse mode) = 0.67 μm
Standard Deviation (fine mode) =1 .697 μm; Standard Deviation (coarse mode) = 1.806 μm

Minimum Radii (fine mode) = 0.00627012 μm; Minimum Radii (coarse mode) = 0.0629802 μm
Maximum Radii (fine mode) = 0.431252 μm; Maximum Radii (coarse mode) = 7.12764 μm

### Axis ratio distribution for the spheroidal dust particles (*Dubovik et al.* 2006)

| Radius | Fractional Weight | Radius | Fractional Weight |
|--------|-------------------|--------|-------------------|
| 0.33490 | 0.0661850 | 1.09540 | 0.0000000 |
| 0.36690 | 0.0650250 | 1.20000 | 0.0000000 |
| 0.40190 | 0.0636350 | 1.31450 | 0.0000000 |
| 0.44030 | 0.0620500 | 1.44000 | 0.0403205 |
| 0.48230 | 0.0587200 | 1.57740 | 0.0429530 |
| 0.52830 | 0.0533500 | 1.72800 | 0.0477625 |
| 0.57870 | 0.0477625 | 1.89290 | 0.0533500 |
| 0.63390 | 0.0429530 | 2.07360 | 0.0587200 |
| 0.69440 | 0.0403205 | 2.27150 | 0.0620500 |
| 0.76070 | 0.0000000 | 2.48832 | 0.0636350 |
| 0.83330 | 0.0000000 | 2.72580 | 0.0650250 |
| 0.91290 | 0.0000000 | 2.98600 | 0.0661850 |
| 1.00000 | 0.0000000 | | |





**Optical and microphysical properties of OMACA Carbonaceous Aerosol Model**

| Model# | Imaginary Index | | | Single-scattering Albedo | | | Absorption Angstrom Exponent | |
|---|---|---|---|---|---|---|---|---|
| | Wavelength in nm | | | Wavelength in nm | | | | |
| | *354* | *388* | *500* | *354* | *388* | *500* | *354-388* | *354-500* |
| 1 | 0.0576 | 0.0480 | 0.0288 | 0.7577 | 0.7806 | 0.8265 | 2.4555 | 2.5080 |
| 2 | 0.0480 | 0.0400 | 0.0240 | 0.7876 | 0.8082 | 0.8486 | 2.5124 | 2.5590 |
| 3 | 0.0360 | 0.0300 | 0.0180 | 0.8288 | 0.8549 | 0.8785 | 2.5935 | 2.6238 |
| 4 | 0.0240 | 0.0200 | 0.0120 | 0.8753 | 0.8879 | 0.9117 | 2.6477 | 2.6821 |
| 5 | 0.0120 | 0.0100 | 0.0060 | 0.9346 | 0.9435 | 0.9603 | 2.8481 | 2.9196 |
| 6 | 0.0060 | 0.0050 | 0.0030 | 0.9646 | 0.9696 | 0.9789 | 2.9343 | 2.9955 |
| 7 | 0.0000 | 0.0000 | 0.0000 | 1.0000 | 1.0000 | 1.0000 | N/A | N/A |

Real refractive index = 1.5

Model # 1 to 3

Mean Radius (fine mode) = 0.080132 μm; Mean Radius (coarse mode) = 0.705495 μm Standard

Deviation (fine mode) = 1.492 μm; Standard Deviation (coarse mode) = 2.075 μm

Minimum Radii (fine mode) = 0.0161708 μm; Minimum Radii (coarse mode) = 0.0380559 μm
Maximum Radii (fine mode) = 0.397083 μm; Maximum Radii (coarse mode) = 13.0788 μm

Model # 4 to 7
Mean Radius (fine mode) = 0.08717 μm; Mean Radius (coarse mode) = 0.567194 μm Standard

Deviation (fine mode) = 1.537 μm; Standard Deviation (coarse mode) = 2.203 μm

Minimum Radii (fine mode) = 0.0156197 μm; Minimum Radii (coarse mode) = 0.0240810 μm
Maximum Radii (fine mode) = 0.486477 μm; Maximum Radii (coarse mode) = 13.3595 μm



**Nodes of OMACA Look-up table**

| Node Parameter | Number of Nodes | Node Values |
|---|---|---|
| ACAOD | 7 | 0.0, 0.1, 0.5, 1.0, 2.5, 4.0, and 6.0 at 500 nm |
| COD | 8 | 2, 5, 10, 20, 30, 40, and 50 (wavelength independent) |
| Solar Zenith Angle | 7 | 0°, 20°, 40°, 60°, 66°, 72°, 80° |
| Viewing Zenith Angle | 14 | 0°, 12°, 18°, 26°, 32°, 36°, 40°, 46°, 50°, 54°, 56°, 60°, 66°, 72° |
| Relative Azimuth Angle | 11 | 0°, 30°, 60°, 90°, 120°, 150°, 160°,165°,170°,175°,180° |
| Surface Pressure Levels | 2 | 1013.25, 800 hPa |
| Aerosol Layer Height | 4 | 3, 4, 5, 6 km |
| Surface Albedo | 5 | 0.00, 0.05, 0.10, 0.15, 0.20 at 354 and 388 nm |




**Content of OMACA HDF-EOS5 Data File**

| SDS Name | Dimensions | Description |
|---|---|---|
| **Geolocation Fields** | | |
| *Longitude* | X,Y | Geodetic longitude of the center part of the pixel (deg) |
| *Latitude* | X,Y | Geodetic latitude of the center part of the pixel (deg) |
| *FoV75Area* | X | Mean Area for 75% Field of View Pixels on the WGS-85 Ellipsoid (km²) |
| *FoV75CornerLongitude* | X,Y,4 | Corner Latitudes for 75% Field of View Pixels on the WGS-85 Ellipsoid (deg) |
| *FoV75CornerLatitude* | X,Y,4 | Corner Latitudes for 75% Field of View Pixels on the WGS-85 Ellipsoid (deg) |
| *SolarZenithAngle* | X,Y | Solar Zenith Angle (deg) |
| *ViewingZenithAngle* | X,Y | Satellite Viewing Zenith Angle (deg) |
| *RelativeAzimuthAngle* | X,Y | Relative Azimuth Angle (deg) SolarZenithAngle + 180 – ViewingZenithAngle |
| *TerrainPressure* | X,Y | Terrain Pressure (mbar) |
| *Time* | Y | Time at the start of each scan (seconds, TAI93) |
| *SecondsInDay* | Y | Seconds of day at start of scan |
| *XTrackQualityFlags* | X,Y | XTrack Quality Flags |
| *GroundPixelQualityFlags* | X,Y | Groud Pixel Quality Flags |
| **Data Fields** | | |
| *AerosolOpticalDepthOverCloud* | X,Y,3 | Aerosol Optical Depth Over Cloud at 354, 388, and 500 nm |
| *AerosolCorrCloudOpticalDepth* | X,Y,3 | Aerosol-corrected Cloud Optical Depth at 354, 388, and 500 nm |
| *ApparentCloudOpticalDepth* | X,Y,3 | Apparent (not corrected for aerosols) Cloud Optical Depth at 354, 388, and 500 nm |
| *FinalAlgorithmFlags* | X,Y | Final Algorithm Flags assigned to each OMACA retrieval |
| *FinalAlgorithmFlags_MieAI* | X,Y | Final Algorithm Flags associated with UV Aerosol Index calculations |
| *CloudOpticalDepth_MieAI* | X,Y | Cloud Optical Depth (388 nm) imported from the OMAERUV AI (Mie) algorithm |
| *CloudFraction_MieAI* | X,Y | Radiative Cloud Fraction (388 nm) imported from the OMAERUV AI (Mie) algorithm |
| *InputSSA354* | X,Y | Aerosol Single-scattering Albedo at 354 nm assumed in the retrieval |
| *InputSSA388* | X,Y | Aerosol Single-scattering Albedo at 388 nm assumed in the retrieval |
| *InputSSA500* | X,Y | Aerosol Single-scattering Albedo at 500 nm assumed in the retrieval |
| *UVAerosolIndex* | X,Y | UV Aerosol Index (354-388 nm pair) imported from the OMAERUV algorithm |
| *NormRadiance* | X,Y,3 | Normalized Radiance at 354, 388, and 500 m |
| *Reflectivity* | X,Y,2 | Reflectivity at 354 and 388 nm |
| *Residue* | X,Y | Residue (354-388 nm pair) |
| *SurfaceAlbedo* | X,Y,2 | Surface Albedo at 354 and 388 nm |
| *FinalAerosolLayerHeight* | X,Y | Final Aerosol Layer Height (km) from the CALIOP-OMI monthly dataset |
| *AIRSL3COvalue* | XY | AIRS Carbon Monoxide L3 data |
| *AerosolType* | X,Y | Aerosol Type assigned to each OMACA retrieval |
| *Wavelength* | 3 | Wavelength of interest (354, 388, 500 nm) |
| *PixelQualityFlags* | X,Y,3 | Pixel Quality Flags for 354, 388, and 500 nm |



**List of Tables**

1. A description of the OMACA algorithm retrieval quality flags.

2. Theoretical simulation of the retrieved ACAOD for given two discrete values of UVAI under different combinations of SSA and ALH assumptions. Threshold values of 0.8 and
1.3 in UVAI correspond to the algorithm quality flags of '0' (best) and '2' (lesser confidence).

3. Theoretical error (%) in ACAOD (388 nm) due to the uncertainty in the assumption of SSA at 388 nm. The
reference value of SSA (388 nm) assumed in the calculation was 0.89; the error in SSA (leftmost column) represents perturbation from the reference value. The cloud optical depth underneath the aerosol layer was assumed to be 10.

4. Theoretical error (%) in ACAOD (388 nm) due to the uncertainty in the assumption of aerosol layer height
(ALH). The reference value of ALH assumed in the calculation was
4.0 km; the error in ALH (leftmost column) represents perturbation from the reference value. The cloud optical depth underneath the aerosol layer was assumed to be 10.

5. Theoretical error (%) in ACAOD (388 nm) due to the uncertainty in the assumption of aerosol Absorption
Angstrom Exponent (AAE). The reference value of AAE (354-388 nm) assumed in the calculation was 2.65; the error in AAE (leftmost column) represents perturbation from this reference value. The cloud optical depth underneath the aerosol layer was assumed to be 10.



**Table 1** A description of the OMACA algorithm retrieval quality flags.

| Algorithm Quality Flags | Observation Conditions | Description |
|---|---|---|
| 0 | UVAI (Mie) >1.3 & LER388>0.25 | Best quality retrievals |
| 1 | 1.3<UVAI (Mie) <4.3& 0.20<LER388<0.25 | Less confidence in the detection of total overcast pixels (Use of high-res sensors is recommended) |
| 2 | 0.8<UVAI (Mie) <1.3 & LER>0.25 | Less confidence in the detection of aerosols above cloud |
| 3 | Solar Zenith Angle> 55° & Scattering Angle <100° & UVAI (Mie) <2  Solar Zenith Angle> 60° & Scattering Angle <130° & UVAI (Mie) <2  Viewing Zenith Angle>55° & Scattering Angle <100° & UVAI (Mie) <2 | Geometry-related artifacts  (not recommended for use) |
| 4 | Snow/Ice Contamination | No retrieval |
| 5 | Solar Zenith Angle above threshold (70°) | No retrieval |
| 7 | Terrain Pressure below threshold (800 hPa). | No retrieval |
| 8 | Cross-track anomaly | No retrieval |





**Table 2** Theoretical simulations of the retrieved ACAOD for given two discrete values of UVAI under different combinations of SSA and ALH assumptions. Threshold values of 0.8 and 1.3 in UVAI correspond to the algorithm quality flags of '0' (best) and '2' (lesser confidence).

| UVAI (Mie) | Aerosol Single-scattering Albedo (388 nm) | | |
|:---:|:---:|:---:|:---:|
| | **SSA388=0.85** | **SSA388=0.90** | **SSA388=0.94** |
| | *ALH 3/4/5 km* | *ALH 3/4/5 km* | *ALH 3/4/5 km* |
| **0.8** | 0.06/0.055/0.05 | 0.18/0.17/0.16 | 0.35/0.33/0.30 |
| **1.3** | 0.22/0.20/0.19 | 0.30/0.28/0.26 | 0.57/0.52/0.49 |





**Table 3** Theoretical error (%) in ACAOD (388 nm) due to the uncertainty in the assumption of SSA at 388 nm. The reference value of SSA (388 nm) assumed in the calculation was 0.89; the error in SSA (leftmost column) represents perturbation from the reference value. The cloud optical depth underneath the aerosol layer was assumed to be 10.

| Error in SSA (388 nm) | Above-cloud AOD (388 nm) | | | | | |
|---|---|---|---|---|---|---|
| | 0.25 | 0.50 | 0.75 | 1.0 | 1.5 | 2.0 |
| **-0.05** | -29.05 | -30.73 | -32.42 | -32.85 | -36.52 | -37.55 |
| **-0.04** | -24.53 | -25.85 | -27.19 | -28.22 | -30.32 | -32.51 |
| **-0.03** | -19.76 | -20.73 | -21.72 | -22.95 | -23.92 | N/R |
| **-0.02** | -14.21 | -14.84 | -15.48 | -16.31 | -16.84 | N/R |
| **-0.01** | -7.71 | -8.02 | -8.33 | -8.73 | -8.96 | N/R |
| **0.00** | 0.00 | 0.00 | 0.00 | 0.00 | 0.00 | 0.00 |
| **0.01** | 11.13 | 11.40 | 11.85 | 13.46 | 15.42 | N/R |
| **0.02** | 24.71 | 25.28 | 26.46 | 29.50 | N/R | N/R |
| **0.03** | 41.60 | 42.80 | 46.25 | N/R | N/R | N/R |
| **0.04** | 63.35 | 66.65 | N/R | N/R | N/R | N/R |
| **0.05** | 90.93 | 98.53 | N/R | N/R | N/R | N/R |



**Table 4** Theoretical error (%) in ACAOD (388 nm) due to the uncertainty in the assumption of aerosol layer height (ALH). The reference value of ALH assumed in the calculation was 4.0 km; the error in ALH (leftmost column) represents perturbation from the reference value. The cloud optical depth underneath the aerosol layer was assumed to be 10.

| Error in ALH (km) | Above-cloud AOD (388 nm) | | | | | |
|:---:|:---:|:---:|:---:|:---:|:---:|:---:|
| | **0.25** | **0.50** | **0.75** | **1.0** | **1.5** | **2.0** |
| **-1.0** | 7.74 | 9.45 | 11.40 | 14.93 | 20.78 | N/R |
| **-0.5** | 3.43 | 4.24 | 5.17 | 6.83 | 9.91 | N/R |
| **0.0** | 0.00 | 0.00 | 0.00 | 0.00 | 0.00 | 0.00 |
| **0.5** | -2.45 | -3.02 | -3.59 | -4.14 | -4.83 | N/R |
| **1.0** | -4.56 | -5.68 | -6.80 | -7.94 | -9.77 | -12.81 |





**Table 5** Theoretical error (%) in ACAOD (388 nm) due to the uncertainty in the assumption of aerosol Absorption Angstrom Exponent (AAE). The reference value of AAE (354-388 nm) assumed in the calculation was 2.65; the error in AAE (leftmost column) represents perturbation from this reference value. The cloud optical depth underneath the aerosol layer was assumed to be 10. Simulations marked with N/R (Not Retrieved) represent the retrieval failure due to the out-of-domain issue.

| Error in AAE (354-388 nm) | Above-cloud AOD (388 nm) | | | | | |
|---|---|---|---|---|---|---|
| | **0.25** | **0.50** | **0.75** | **1.0** | **1.5** | **2.0** |
| **-1.5** | 83.17 | 96.36 | 102.80 | N/R | N/R | N/R |
| **-1.0** | 48.42 | 54.12 | 58.50 | N/R | N/R | N/R |
| **-0.5** | 20.80 | 22.14 | 24.38 | 26.60 | N/R | N/R |
| **0.0** | 0.00 | 0.00 | 0.00 | 0.00 | 0.00 | 0.00 |
| **0.5** | -11.67 | -12.33 | -12.84 | -13.40 | -13.54 | -14.50 |
| **1.0** | -21.69 | -22.96 | -24.00 | -24.69 | -25.65 | -27.69 |
| **1.5** | -30.69 | -32.54 | -34.12 | -34.68 | -36.89 | -40.09 |









**Figure 10** Time-series of regional, monthly mean aerosol-corrected (left y-axis, color: red) COD (388 nm) derived using a 12-year long OMACA record. The y-axis on the right side (color: blue) depicts a time-series of the % difference between simultaneous retrievals of apparent/non-corrected and aerosol-corrected COD for the same period and regions. Only anomaly-free observations of OMI (Rows 1 to 23) were used in the calculations.

**Figure 11** Percent difference in COD (aerosol-corrected minus non-corrected) as a function of above-cloud absorption AOD (388 nm) for the smoke (left panel) and dust (right panel) dominated regions. Different color codes represent the relationship for a discrete range of aerosol-corrected COD.





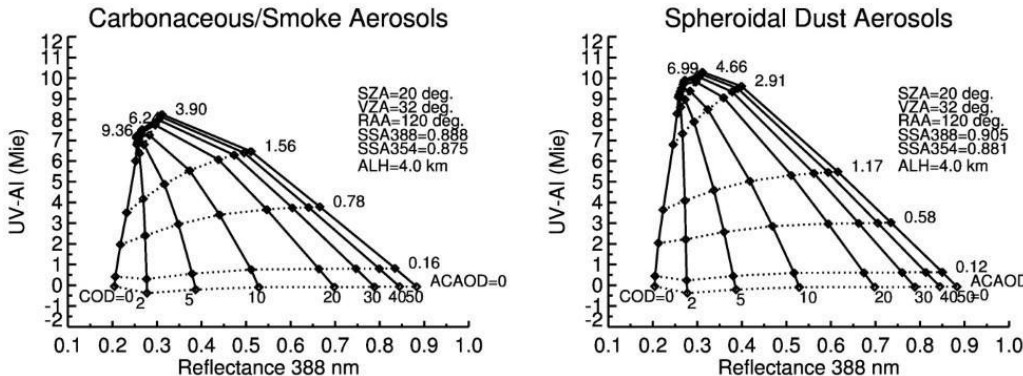

**Figure 1** Simulation of UVAI (y-axis) versus Reflectance at 388 nm (x-axis) for the different pairs of ACAOD and COD (both at 388 nm) for the carbonaceous (left) and spheroidal shape dust aerosols (right). Values of ACAOD and COD depicted in the figure correspond to the 388 nm wavelength. The shown 2-D diagram forms the retrieval domain in which the observations from OMI are fitted and related to a pair of ACAOD and COD.





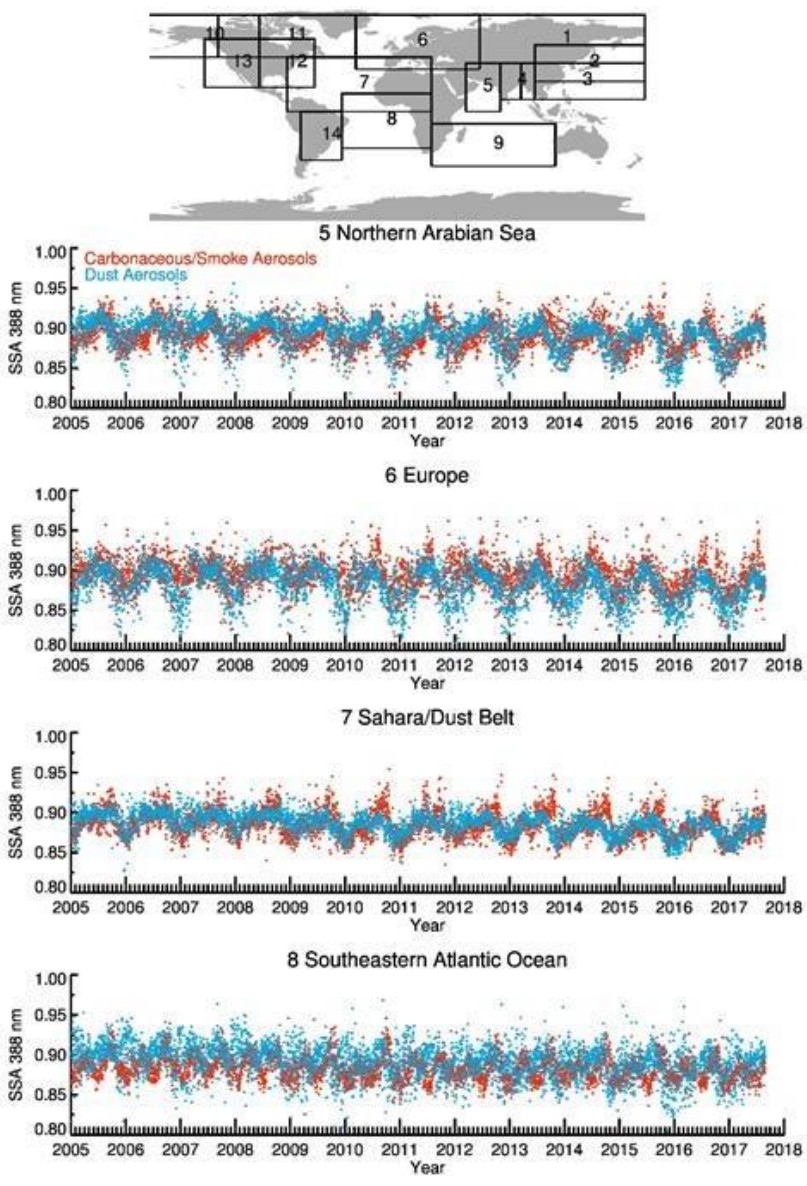

**Figure 2** (Top) Geographical boundaries of select 14 regions considered for calculating corresponding regional
values of SSA (388 nm). (Remaining panels) Regional mean UVAI weighted cloud-free SSA (388 nm) for
carbonaceous, and dust aerosols for the four selected regions (numbered 5, 6, 7, and 8 in the top panel) derived using
OMI/OMAERUV operational (version 1.8.9.1) L2 cloud-free SSA retrievals.





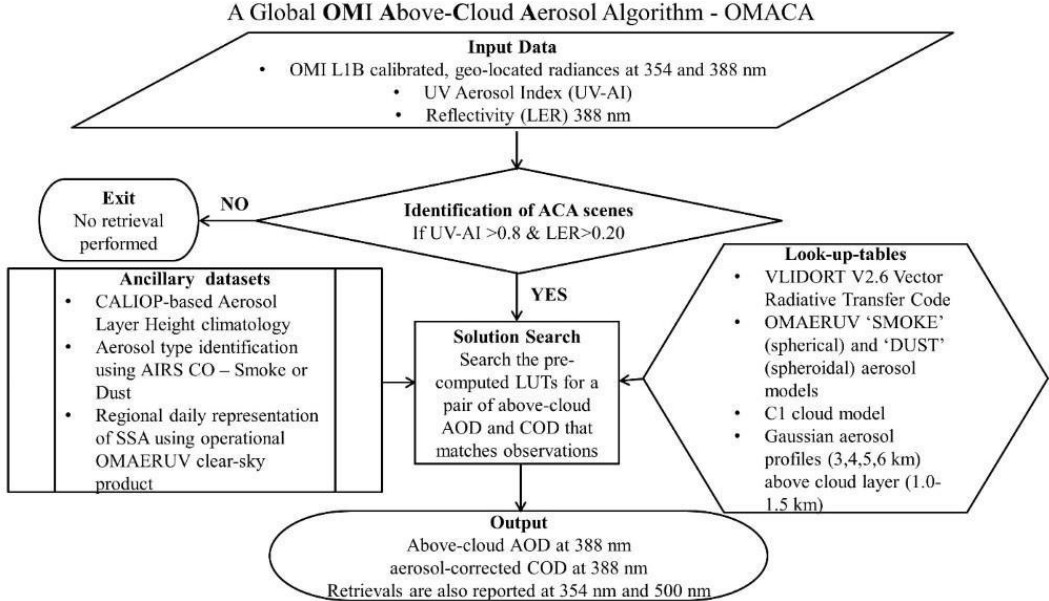

**Figure 3** A General flowchart of the OMACA algorithm.







**Figure 4** Comparison of coincident and collocated spectral ACAODs measured and retrieved from HSRL-2 and OMI/OMACA for a total of seven ER-2 flights operated during ORACLES phase I operation over the Southeastern Atlantic Ocean in August-September of 2016. ΔT represents the absolute time difference (in hours) between the OMI overpass and HSRL-2 measurements.



**Figure 5** The monthly mean distribution of cloudy-sky frequency of occurrence of above-cloud absorbing aerosols
deduced from the 12-year record (2005-2016) of OMI's near-UV observations.




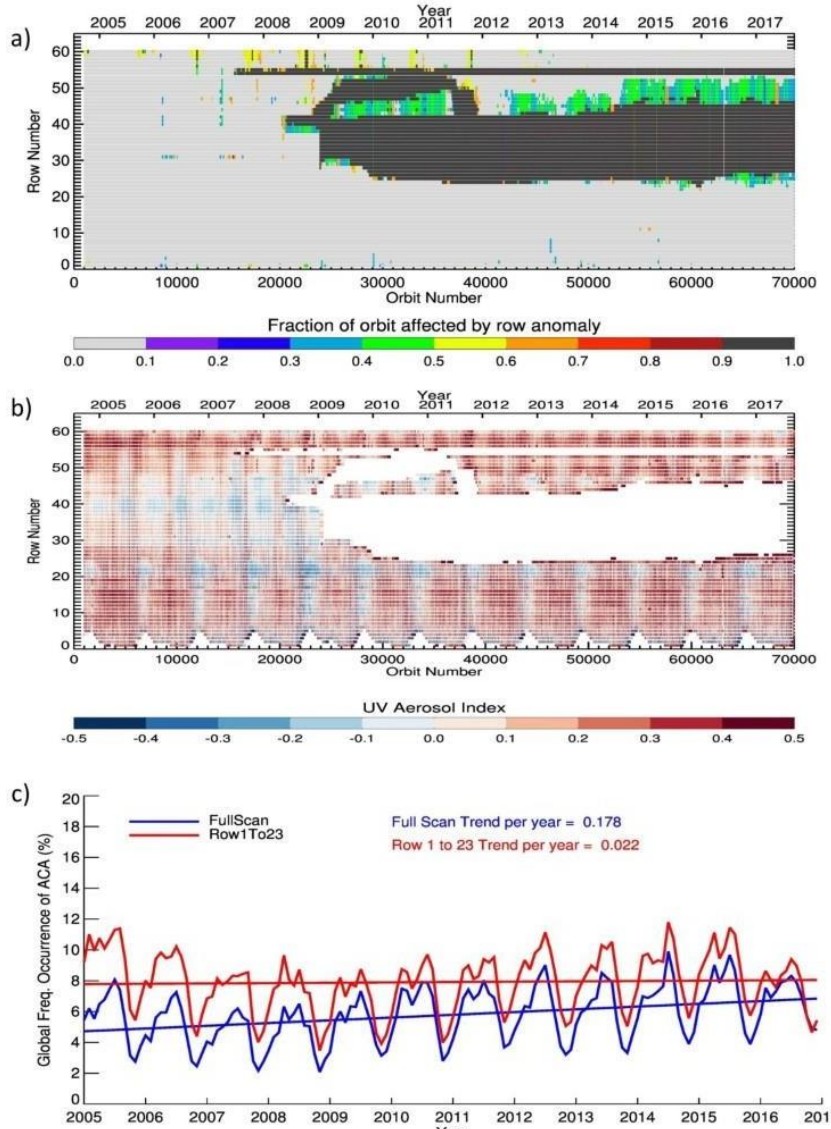

**Figure 6** (a) Chart showing the time evolution of OMI row anomaly and (b) cloudy-sky (LER388>0.25) UVAI. (c) Monthly time-series of the global, cloudy-sky frequency of occurrence (in % with respect to the total cloudy-sky observations) of absorbing aerosols above cloud derived using OMI full scan (Rows 1 to 60) (blue) and anomaly-free observations (Rows 1 to 23) (red). Solid lines represent the linear regression fits to the respective time-series data.



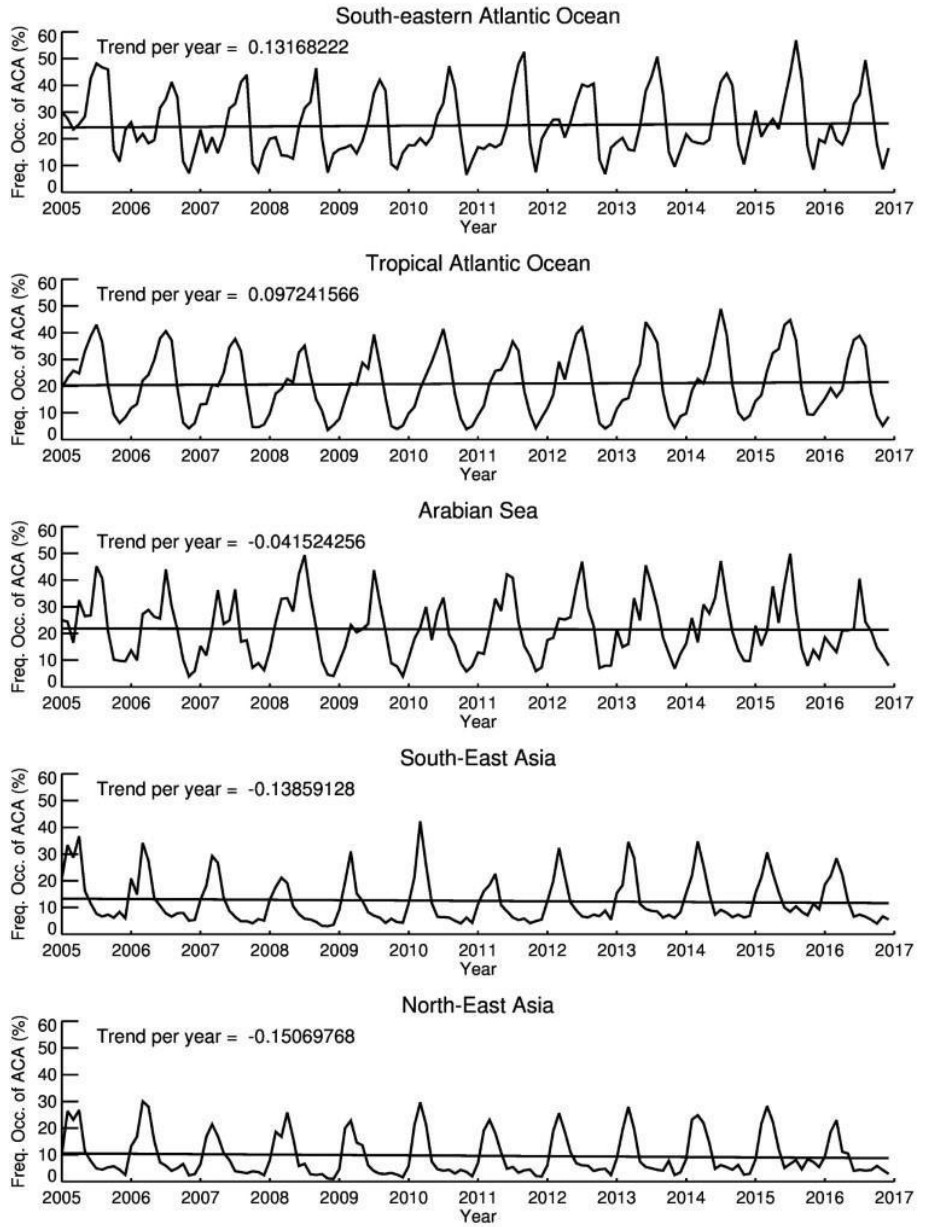

**Figure 7** Monthly time-series of the regional cloudy-sky frequency of occurrence (in % with respect to the total cloudy-sky observations) of absorbing aerosols above cloud derived using OMI anomaly-free observations (Rows 1 to 23). Solid lines represent the linear fit to the respective time-series data.





**Figure 8** Global distribution of monthly mean above-cloud AOD (388 nm) deduced from the 12-year (2005-2016)

5      OMI observations.



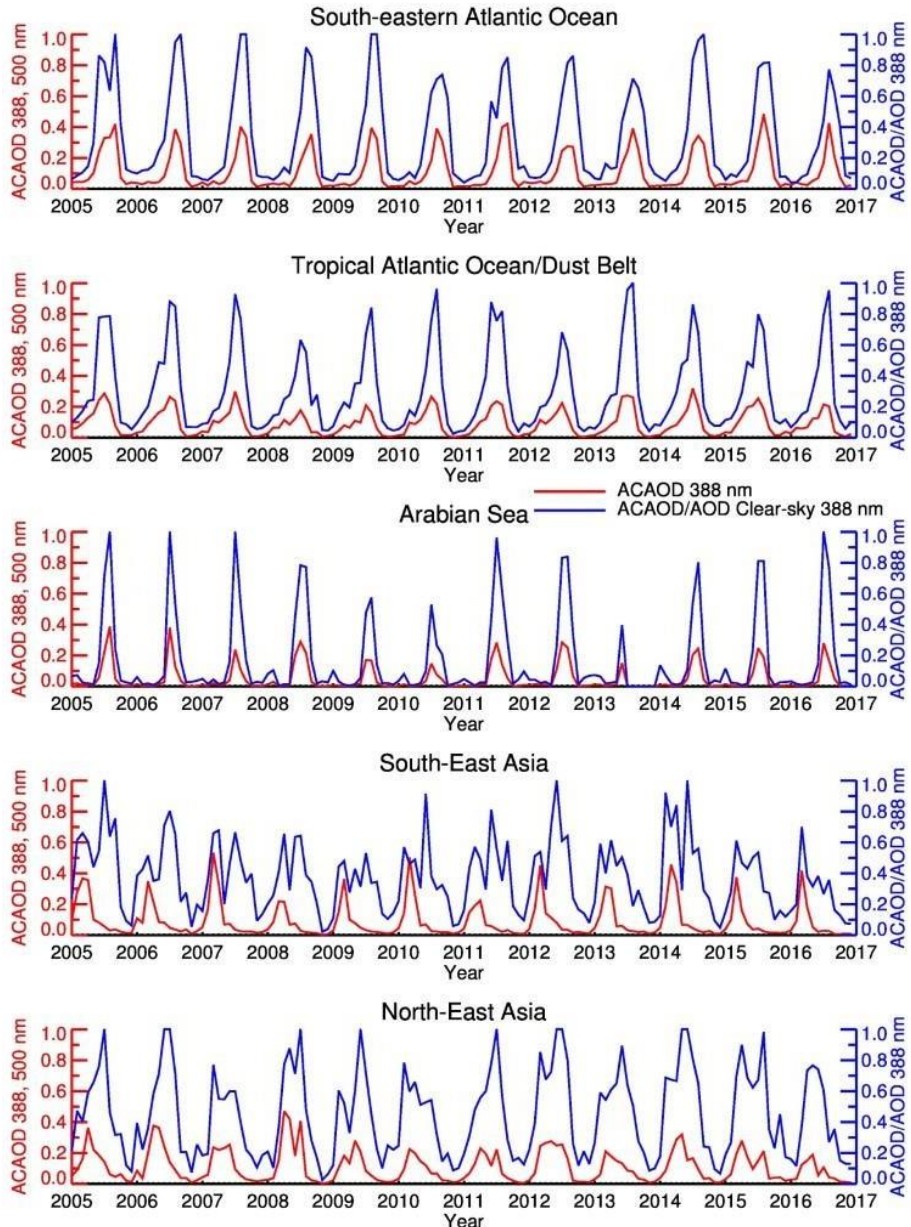

**Figure 9** Time-series of regional monthly mean above-cloud AOD at 388 nm (left y-axis, color: red) and the ratio of above-cloud AOD and cloud-free AOD (right y-axis, color: blue), both at 388 nm, derived using OMACA and OMAERUV products, respectively.





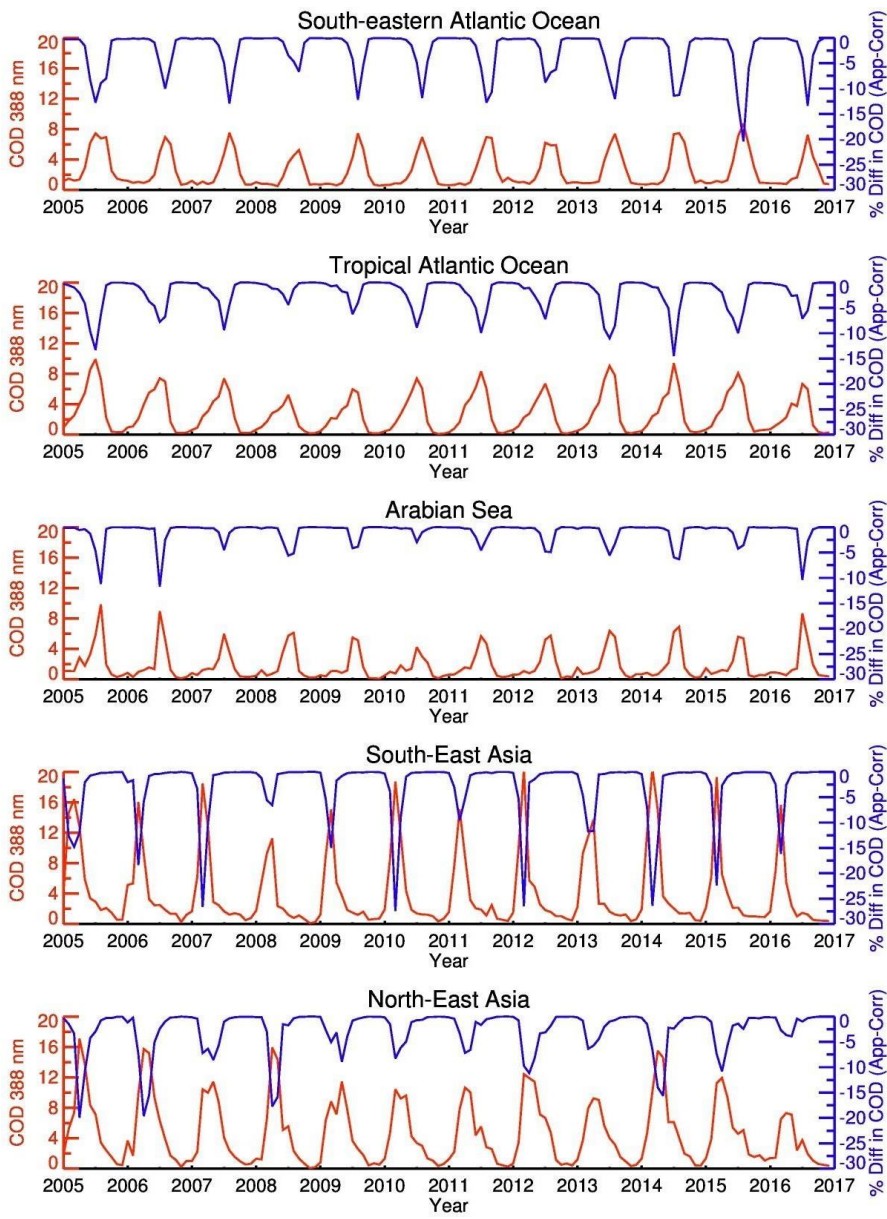

**Figure 10** Time-series of regional, monthly mean aerosol-corrected (left y-axis, color: red) COD (388 nm) derived using a 12-year long OMACA record. The y-axis on the right side (color: blue) depicts a time-series of the % difference between simultaneous retrievals of apparent/non-corrected and aerosol-corrected COD for the same period and regions. Only anomaly-free observations of OMI (Rows 1 to 23) were used in the calculations.





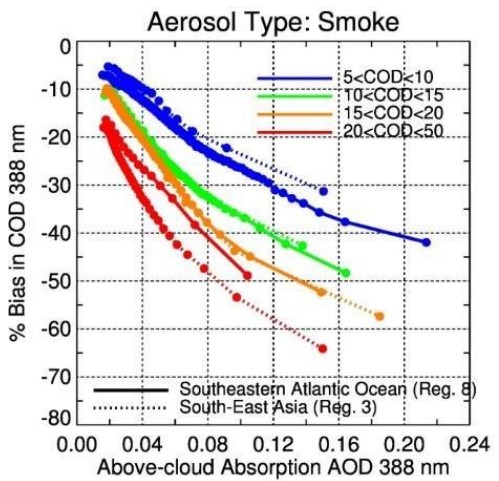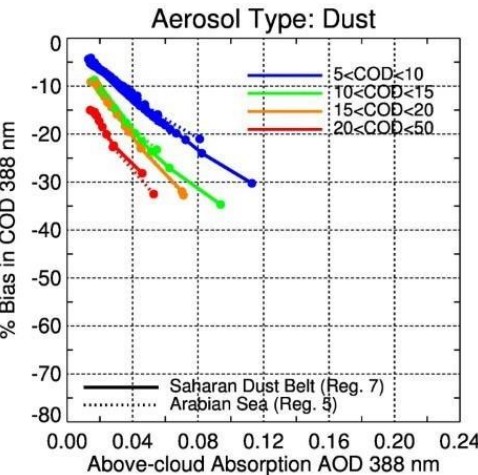

**Figure 11** Percent difference in COD (aerosol-corrected minus non-corrected) as a function of above-cloud absorption AOD (388 nm) for the smoke (left panel) and dust (right panel) dominated regions. Different color codes represent the relationship for a discrete range of aerosol-corrected COD.