# Peer review of "A 12-Year Long Global Record of Optical Depth of Absorbing Aerosols above the Clouds Derived from OMI/OMACA Algorithm"

_Atmospheric Measurement Techniques, 2018_

## Referee Comment (RC1) · Anonymous Referee #1 · 4 Jul 2018

In this paper, the authors discussed a new above cloud absorbing aerosol optical depth product. Uncertainty analysis and preliminary validations are provided. Spatial distributions of ACAOD as well as 12 year trends are also included. The authors did a nice job presenting their new ACAOD product, still I have some comments that I hope the authors could address.

Both ACAOD and aerosol corrected COD are derived from their retrieval algorithm. However, no validation effort is provided for the derived aerosol corrected COD values. I understand that this is a paper that focuses on ACAOD, but the authors shall at least discuss how the uncertainties in the derived aerosol corrected COD would affect the

derived ACAOD and vise versa.

For this product to have a board user community, and especially for modelers, the uncertainties in retrieved properties (such as aerosol corrected COD and ACAOD) shall be provided at the individual retrieval level. Uncertainties are not included in the current data fields as shown in Appendix I. Given the uncertainty in SSA of +-0.03, an uncertainty in ACAOD could be introduced on the order of 20-50% (Table 3). I wonder how much the uncertainties in their regional and global time series of ACAOD are attributed to uncertainties in SSA, or are a direct reflection of temporal variations in SSA. Other comments:

Page 5, lines 32-33, I am not really sure what the authors mean by this sentence "These two components the OMACA algorithm is identical to the ones adopted in the operational cloud-free OMI/OMAERUV two-channel algorithm."

Page 5, line 24, The ALH dataset ,which was derived using 30 months of collocated CALIOP and OMI data, is used for aerosol vertical profiles. Was this ALH dataset derived using aerosol above cloud scenes only? If the ALH dataset was derived using cloud free scenes, how representative is the ALH dataset for aerosol above cloud cases?

Page 6, lines 29-30, "Observations of aerosols above cloud found outside the boundaries of these 14 pre-selected regions are assigned a fixed SSA of 0.89 and 0.9 for the smoke and dust aerosol types, respectively." Justifications or references are needed for values mentioned here.

Page 8, lines 23-25, "Notice that the current OMACA product does not use the OMMYDCLD product while making above-cloud aerosol retrieval. Instead, we use the information on the geometric cloud fraction derived from OMMYCLD in the post-retrieval analysis." I wonder why the OMMYDCLD product is not used in the retrieval process. The authors seem to use LER > 0.2 to distinguish clear from cloudy scenes. But wouldn't the use of OMMYCLD result in a more accurate estimation of cloud coverage

over a given scene? How do the authors deal with partially cloudy scenes?

Page 10, lines 20-21, "53% (AOD>0.7, UVAI>1.0) of the total OMAERUV-AERONET SSA (440 nm) retrievals are found to agree within their estimated uncertainties of ±0.03." This means 47% (AOD>0.7, UVAI>1.0) of SSA retrievals are outside of the uncertainty range of +-0.03. I am not sure how the authors could come up with this statement "Therefore, we expect that the above-cloud SSA values assigned in the OMACA algorithm over different regions should be accurate within ±0.03."

---

## Referee Comment (RC2) · Z. Zhang (Referee) · 8 Jul 2018

General Comments: This paper documents the above-cloud aerosol retrieval algorithm based on the OMI observations (OMIACA) and the corresponding 12-year climate data record of above-cloud aerosol optical depth from this algorithm. The paper starts with the description of the theoretical base, implementation and uncertainty analysis of the algorithm, followed by a climatology study of the geographical distribution of above-cloud aerosols, their occurrence frequencies and their AOD trend.

The OMACA product, along with several other products, provides the much-needed observations for studying the above-cloud aerosols and their interactions with clouds

and radiation. This paper provides the documentation of the algorithm that enabled the OMACA product and therefore is an important contribution to the literature.

This paper very well was written and organized. Its topic is suitable for AMT. I only have some minor comments/suggestions for the authors to consider to further improve the paper.

Specific comments:

Introduction: a couple of important papers in the field of above-cloud aerosol studies are missing in the introduction. They should be cited to give the readers a more comprehensive and complete overview of the field. o Devasthale and Thomas [2011] is among the first to study the occurrence of above-cloud aerosols.

o Zhang et al. [2016] further studied the occurrence frequencies of different types of above-cloud aerosols over the globe using 8 years of CALIOP observations. They also derived the shortwave direct radiative effects of above-cloud aerosols based on the collocated observations of CALIOP ACA AOD and MODIS COD.

o Min and Zhang [2014] showed that the direct radiative effects of ACA also depend on the diurnal cycle of cloud. This study should be cited when discussing the factors influencing the DRE of ACA (i.e., around line 14)

o A recent study by Lu et al. [2018] showed that the entrained ACA from cloud top can significantly influence the cloud microphysics and actually brightens the clouds in the SE Atlantic region. This paper can be cited with Wilcox 2012 on the importance of ACA.

I think a brief overview of the existing above-cloud aerosol retrievals algorithms for the passive sensors will give the readers a "big picture" and understand the relative position of this study. In particular, as the authors are aware, the following algorithms have been developed for POLDER and MODIS

o Waquet et al. [2013b] and Waquet et al. [2013a] described a novel algorithm for

retrieving the ACA properties from the POLDER observations.

o Meyer et al. [2015] developed a method for retrieving the ACA-AOD and COD below simultaneously from MODIS observations.

o Sayer et al. [2016] extended the "deep blue" aerosol retrieval algorithm to the above-cloud conditions.

Aerosol type identification (section 2.2.2.1): This part is very important, Because the retrieval algorithm uses different optical properties for different type of aerosols, i.e., dust or smoke. A misclassification can cause retrieval errors and uncertainties. However, the description of this paper is very brief. Some more details need to be added with proper references. For example, it should be mentioned whether and how the identification scheme is validated or evaluated. Has it been compared with CALIOP aerosol subtypes? Why different threshold of CO is used for northern and southern hemispheres?

• Single scattering albedo (section 2.2.2.1): the SSA is extremely important for ACA retrieval and DRE. I hope the information of aerosol type and the corresponding SSA will be part of the OMIACA so the users can use it in a consistent way with the AOD product.

• Look-up-tables (section 2.2.24): How is cloud effective radius considered in the LUT? Is it assumed as a constant? Note that the assumption of CER could have impacts on the COD retrieval. Some discussion is needed to clarify this.

• Partly cloudy pixels: the footprint size of OMI is 13x24 km (338km2). At this scale, there cloud be a lot of partly cloudy pixels. One of my main questions/concerns is about the treatment of the partly cloudy OMI pixels. It seems to me the OMIACA algorithm is applied to both overcast pixels and partly cloudy pixels, correct? How is the subpixel cloud fraction determined? How are the partly cloudy pixels treated in the LUT and radiative transfer simulations? How is the UVAI of clear-sky part of the partly cloudy

pixel different from that of the cloudy part, and what is the meaning of the "observed" UVAI for the partly cloudy pixel? I would strongly recommend the authors to add a separate and dedicated sub-section to discuss the treatment of partly cloudy pixels in the OMIACA algorithm.

• Sub-pixel COD variation: A related question is whether and how the algorithm accounts for the subpixel COD variation. What is the physical meaning of the "retrieved COD"? Is it a simple mathematical mean or some kind of weighted mean?

• Spatial distribution of ACA: As mentioned above Zhang et al. (2016) studied the global distribution of different types of ACA. Actually, Figure 5 agrees reasonably well with the Figure 2 of Zhang et al. (2016). In addition, Zhang et al. (2016) also found significant amount of ACA over the north pacific due to the Asian dust and pollution. This study should be cited here and discussed.

• Figure 5 shows some ACA over the Southern Ocean in January and February. Is this true or some retrieval artifact?

• It is interesting to see that the FOACA in Figure 5 and the ACA AOD in Figure 8 are highly correlated. Is this a coincidence or there may be some real connection between them? One could argue that the region with high FOACA does not necessarily have larger ACA AOD. Do you agree?

Devasthale, A., and M. A. Thomas (2011), A global survey of aerosol-liquid water cloud overlap based on four years of CALIPSO-CALIOP data, Atmospheric Chemistry and Physics, 11(3), 1143–1154, doi:10.5194/acp-11-1143-2011.

Lu, Z., X. Liu, Z. Zhang, C. Zhao, K. Meyer, C. Rajapakshe, C. Wu, Z. Yang, and J. E. Penner (2018), Biomass smoke from southern Africa can significantly enhance the brightness of stratocumulus over the southeastern Atlantic Ocean, PNAS, 115(12), 201713703–2929, doi:10.1073/pnas.1713703115.

Meyer, K., S. Platnick, and Z. Zhang (2015), Simultaneously inferring above‐cloud

absorbing aerosol optical thickness and underlying liquid phase cloud optical and microphysical properties using MODIS, Journal of Geophysical Research-Atmospheres, 120(11), 5524–5547, doi:10.1002/2015JD023128.

Min, M., and Z. Zhang (2014), On the influence of cloud fraction diurnal cycle and sub-grid cloud optical thickness variability on all-sky direct aerosol radiative forcing, Journal of Quantitative Spectroscopy and Radiative Transfer, 142 IS -, 25–36, doi:10.1016/j.jqsrt.2014.03.014.

Sayer, A. M., N. C. Hsu, C. Bettenhausen, J. Lee, J. Redemann, B. Schmid, and Y. Shinozuka (2016), Extending "Deep Blue" aerosol retrieval coverage to cases of absorbing aerosols above clouds: Sensitivity analysis and first case studies, Journal of Geophysical Research-Atmospheres, 121(9), 4830–4854, doi:10.1002/2015JD024729.

Waquet, F. et al. (2013a), Retrieval of aerosol microphysical and optical properties above liquid clouds from POLDER/PARASOL polarization measurements, Atmos. Meas. Tech., 6(4), 991–1016, doi:10.5194/amt-6-991-2013.

Waquet, F., F. Peers, F. Ducos, P. Goloub, S. Platnick, J. Riedi, D. Tanre, and F. Thieuleux (2013b), Global analysis of aerosol properties above clouds, Geophys Res Lett, 40(21), 5809–5814, doi:10.1002/2013GL057482.

Zhang, Z., K. Meyer, H. Yu, S. Platnick, P. Colarco, Z. Liu, and L. Oreopoulos (2016), Shortwave direct radiative effects of above-cloud aerosols over global oceans derived from 8 years of CALIOP and MODIS observations, ACP, 16(5), 2877–2900, doi:10.5194/acpd-15-26357-2015.

---

## Author Comment (AC1) · 31 Aug 2018

The authors are thanking the anonymous reviewer for offering constructive comments, which have helped us refine the content of our manuscript.

Following is the response to each comment and suggestion made by the reviewer.

RC: Referee Comment AC: Author's Response

RC: Both ACAOD and aerosol corrected COD are derived from their retrieval algorithm. However, no validation effort is provided for the derived aerosol corrected COD values. I understand that this is a paper that focuses on ACAOD, but the authors shall at least

discuss how the uncertainties in the derived aerosol corrected COD would affect the derived ACAOD and vise versa.

AR: The near-UV 'color ratio' technique employed in the OMACA algorithm retrieves both ACAOD and aerosol-corrected COD simultaneously. The two pieces of information, i.e., UVAI and reflectance at 388 nm from OMI, allows us retrieving two quantities as reflected in the 2D retrieval diagram shown in Figure 1. While the primary focus of the paper is to highlight the ACAOD product, its spatial-temporal distribution, and initial validation against ORACLES/HSRL-2 observations, we are working with the OR-ACLES team to perform a detailed validation of the OMACA product, both ACAOD and COD, using airborne in situ and remote sensing measurements. The results of the validation analysis will be covered in a dedicated follow-up publication.

Section 4 Preliminary Validation is now updated with our intention of a follow-up publication on a detailed validation of the OMACA product.

We don't expect to see an interdependency of ACAOD and aerosol-corrected COD retrievals as both quantities are derived simultaneously that explain the two-channel TOA measurements. The inversion philosophy is quite analogous to that adopted in the standard MODIS cloud retrievals where the TOA signal at 0.86 $\mu$m is largely responsible for COD magnitudes, and 2.1 $\mu$m is for the effective radius. In the present case, the reflectance at 388 nm with an aerosol correction determines the retrieved values of COD and magnitudes of the observed UVAI drives the ACAOD retrievals. So, the assumptions about aerosol and cloud models made in the inversions are the ones that determine the actual uncertainties in both retrievals.

RC: For this product to have a board user community, and especially for modelers, the uncertainties in retrieved properties (such as aerosol corrected COD and ACAOD) shall be provided at the individual retrieval level. Uncertainties are not included in the current data fields as shown in Appendix I. Given the uncertainty in SSA of +-0.03, an uncertainty in ACAOD could be introduced on the order of 20-50% (Table 3). I

wonder how much the uncertainties in their regional and global time series of ACAOD are attributed to uncertainties in SSA, or are a direct reflection of temporal variations in SSA.

AR: Reviewer has correctly pointed out here that the uncertainties in the retrieved properties aren't provided at the individual retrieval level. This is a complex task due to the very nature of uncertainty in ACAOD/COD that depends on multiple assumptions made in the inversion. Errors in the ACAOD retrievals resulting from the uncertainty in each major assumption, i.e., SSA, ALH, and AAE, have already been tabulated in Table 3, 4, and 5. The uncertainty in these assumptions can vary on both sides, i.e., underestimation and overestimation, from the assumed state. Moreover, individual uncertainties can also be of opposite signs leading to the partial cancellation of errors in the retrievals. In this situation, it is hard to estimate actual errors in the ACA retrievals especially when the true state of the atmosphere is unknown to the algorithm.

Looking at the potential use of the OMACA product in the numerical models, we will consider including the pixel-level retrieval uncertainties in the next upgrade on the basis of the sensitivity of ACAOD/COD to these assumptions individually, such as presented in Table 3, 4, and 5.

We assume here that uncertainty in the SSA is random, i.e., distributed on both sides, positive and negative. An OMI-AERONET comparison figure shown to address one of the following comments show that OMI SSAs are spread over both sides of collocated AERONET inversion. Given this assumption, it is expected that the resulting errors in ACAOD/COD retrievals are subjected to the partial cancellation on a monthly regional scale. Moreover, the daily regional time-series of SSA displayed in Figure 2 do not show any apparent long-term trend; rather it demonstrates the known seasonal variations in aerosol absorption properties for both smoke and dust.

RC: Page 5, lines 32-33, I am not really sure what the authors mean by this sentence "These two components the OMACA algorithm is identical to the ones adopted in the

operational cloud-free OMI/OMAERUV two-channel algorithm."

AR: This sentence refers to the Aerosol Layer Height and Surface Albedo datasets—both are directly adopted from the OMAERUV cloud-free algorithm. Not only that, the aerosol type identification scheme and aerosol models (microphysical and optical properties) are also identical to the ones designed for the OMAERUV.

RC: Page 5, line 24, The ALH dataset, which was derived using 30 months of collocated CALIOP and OMI data, is used for aerosol vertical profiles. Was this ALH dataset derived using aerosol above cloud scenes only? If the ALH dataset was derived using cloud-free scenes, how representative is the ALH dataset for aerosol above cloud cases?

AR: The OMI-CALIOP climatology of ALH was derived using mostly clear-sky observations in both datasets with maximum LER in OMI dataset restricted to 0.25. The threshold in LER (0.25) largely avoided ACA scenes that are assigned with the best quality OMACA retrievals (Table 1).

The time-series of fractional AOD shown in Figure 9 (right-hand axis in blue ink) of the original manuscript demonstrated that a large fraction of the total column AOD, about 80%-100%, was retrieved as aerosols above the clouds during the seasonal biomass burning and dust episodes. Moreover, the transport of aerosols over the ACA hotspot regions is known to occur above the boundary layer and between the altitudes 3-6 km, as observed from CALIOP lidar. Note that the minimum (maximum) ALH assumed in the OMACA was fixed to 3.0 km (6.0 km) even if the gridded ALH climatology dataset is assigned with lower (higher) layer height. For these reasons, we expect that the OMI-CALIOP ALH database derived from cloud-free observations over the ACA regions is also a representative for scenes with aerosols above the cloud. The remaining uncertainty in ALH would translate into the corresponding error in ACAOD already presented in Table 3.

This description has been added in the revision (section 3 Uncertainty Estimates)

RC: Page 6, lines 29-30, "Observations of aerosols above cloud found outside the boundaries of these 14 pre-selected regions are assigned a fixed SSA of 0.89 and 0.9 for the smoke and dust aerosol types, respectively." Justifications or references are needed for values mentioned here.

AR: The SSA values of 0.89 (smoke) and 0.9 (dust) prescribed for regions outside the 14 pre-selected regions are merely our assumptions. These values correspond to the aerosol model having a moderate level of absorption for both aerosol types (see Appendix I, aerosol models). We emphasize here that though the OMACA is a global product, it was primarily designed to capture ACA events over major and some minor regions of the world. The selection of regional boundaries was made according to the monthly and seasonal ACA frequency of occurrence maps shown in Figure 5. The pre-selected regions adequately encompass the areas of frequent aerosol-cloud overlaps.

This description has been further clarified in the revised paper (section 2.2.2.3)

RC: Page 8, lines 23-25, "Notice that the current OMACA product does not use the OMMYDCLD product while making above-cloud aerosol retrieval. Instead, we use the information on the geometric cloud fraction derived from OMMYCLD in the post-retrieval analysis." I wonder why the OMMYDCLD product is not used in the retrieval process. The authors seem to use LER > 0.2 to distinguish clear from cloudy scenes. But wouldn't the use of OMMYCLD result in a more accurate estimation of cloud coverage over a given scene? How do the authors deal with partially cloudy scenes?

AR: As the reviewer has correctly mentioned here, the OMMYDCLD product hasn't been used while deriving the OMACA product. However, we positively consider integrating the OMMYDCLD product into the OMACA algorithm in the near-future upgrade. This will help us to better understand OMI sub-pixel cloud variability and its impact on the retrievals.

However, we have used the OMMYDCLD product in the post-retrieval analysis to derive all results presented in this paper. Using the information on the geometric cloud fraction

calculated from OMMYDCLD, we adopted a threshold of 0.50 and 0.75 to carry out the frequency of ACA and ACAOD/COD analyses, respectively (section 2.2.3).

The minimum LER threshold for detecting cloudy pixels is chosen to be 0.2. The retrieval associated with LER range 0.20-0.25 are flagged as of lower quality due to the chances of encountering partly cloudy-pixels for these retrievals (see Table 1). However, retrievals with LER>0.25 are considered to be more reliable (QFlag=0, best quality) owing to the increased probability of detecting fully overcast pixels. Currently, the OMACA algorithm treats each identified cloudy pixel as a fully overcast scene, even if it is partly cloudy as per the OMMYDCLD product.

RC: Page 10, lines 20-21, "53% (AOD>0.7, UVAI>1.0) of the total OMAERUV-AERONET SSA (440 nm) retrievals are found to agree within their estimated uncertainties of ±0.03." This means 47% (AOD>0.7, UVAI>1.0) of SSA retrievals are outside of the uncertainty range of +-0.03. I am not sure how the authors could come up with this statement "Therefore, we expect that the above-cloud SSA values assigned in the OMACA algorithm over different regions should be accurate within ±0.03."

AR: The statements referring to the statistics of OMI versus AERONET SSA comparison were derived from Jethva et al. [2014] paper based on the earlier OMAERUV dataset (version 1.4.2 released in 2012). Since then, the OMAERUV algorithm has been upgraded with several major changes, including better treatment of dust particles assuming realistic spheroidal shape distribution, accounting for angular scattering effects of clouds in the calculation of UV Aerosol Index, use of new minimum surface LER dataset using synergy of multi-year OMI and MODIS observations, and updated cloud screening and retrieval flagging scheme.

The regional, daily SSA dataset used in the OMACA product has been derived from this latest version of the OMAERUV (version 1.8.9.1) product released in 2017. As already described in the manuscript (section 2.2.2.3), we use UVAI-weighted averages of SSA for smoke and dust aerosol types using retrievals with measured UVAI>0.8 assuring

that only high-quality retrievals are used in these calculations. The figure shown here compares the SSA derived from the latest OMAERUV product and the AERONET Level 2 dataset over global AERONET sites (unpublished). The comparative analysis demonstrates that the agreement between the two independent sets of SSA improves significantly at higher values of UVAI. Quantitatively, about 59% (83%), 65% (88%), and 72% (91%) of the matchups are found to be within the expected limits of $\pm 0.03$ ($\pm 0.05$) difference given the observed range of UVAI>0.8, >1.5, and >2.0, respectively.

The reason for adopting an UVAI-weighted scheme precisely reflects the fact that the agreement between OMI and AERONET SSA improves at higher aerosol loading/absorption providing increased confidence in the satellite retrievals.

In the revision, we have modified the description of the OMI-AERONET SSA statistics according to the new comparison results discussed above, since it is consistent with the SSA dataset used in the OMACA algorithm.

[Figure]

[Figure]

**Figure.** Comparison of OMI-retrieved cloud-free SSA (y-axis) against that of AERONET (x-axis) over globally distributed network for OMI UVAI measurements >0.8 (top), >1.5 (middle), and >2.0 (bottom). Only smoke (red) and dust (blue) aerosols types (relevant to OMACA product) as identified by the OMAERUV algorithm are considered. The statistics of the comparison are shown within each plot.

**Fig. 1.**

---

## Author Comment (AC2) · 31 Aug 2018

The authors are thanking the anonymous reviewer for offering constructive comments, which have helped us improve the content of our manuscript.

Following is the one-to-one response to each comment made by the reviewer.

RC: Referee Comment AR: Author's Response

RC: Introduction: a couple of important papers in the field of above-cloud aerosol studies are missing in the introduction. They should be cited to give the readers a more comprehensive and complete overview of the field. Devasthale and Thomas [2011] is

among the first to study the occurrence of above-cloud aerosols.

AR: We realized that we missed these citations in the earlier version of the paper. Relevant papers, as suggested by the referee, are now added/cited at appropriate places in the revised paper.

• Zhang et al. (2016) cited for direct radiative effects of aerosols above cloud. • Min and Zhang (2014) cited along with the statement listing several parameters affecting radiative effects. • Devasthale and Thomas (2011) cited along with the statement "Such situations are commonly observed from satellites over several oceanic and continental regions.." • Devasthale and Thomas (2011) and Zhang et al. (2016) cited and mentioned in section 5.1 describing the results on frequency occurrence of ACA. • Lu et al. [2018] now cited along with Wilcox [2012].

RC: I think a brief overview of the existing above-cloud aerosol retrievals algorithms for the passive sensors will give the readers a "big picture" and understand the relative position of this study. In particular, as the authors are aware, the following algorithms have been developed for POLDER and MODIS

AR: A discussion on the existing state of the active and passive-sensor based ACA algorithms are now added to the Introduction (2nd paragraph).

RC: Aerosol type identification (section 2.2.2.1): This part is very important, Because the retrieval algorithm uses different optical properties for different type of aerosols, i.e., dust or smoke. A misclassification can cause retrieval errors and uncertainties. However, the description of this paper is very brief. Some more details need to be added with proper references. For example, it should be mentioned whether and how the identification scheme is validated or evaluated. Has it been compared with CALIOP aerosol subtypes? Why different threshold of CO is used for northern and southern hemispheres?

AR: The aerosol type identification scheme in the OMACA has been directly adopted

from the cloud-free OMAERUV algorithm. The scheme uses real-time observation of AIRS CO information in conjunction with OMI UVAI to discern the carbonaceous smoke aerosols from mineral dust, which otherwise not possible to detect using only near-UV measurements. The use of CO measurements also enables the identification of high levels of boundary layer pollution undetectable by near-UV observations alone. Since Torres et al. [2013] adequately describes the methodology and implementation of the scheme within OMAERUV, we didn't include a lengthy discussion on this topic in the present manuscript.

The different threshold values of CO in Northern and Southern hemispheres correspond to the average of AIRS CO climatological annual minima over major biomass burning/boreal fire activity regions. These values are 2.2x1018 in the Northern Hemisphere (NH) and 1.8x1018 for the Southern Hemisphere (SH), based on Yurganov et al. [2008, 2010]. The presence of carbonaceous aerosols is assumed if AI $\geq$ AI threshold (0.8) and CO $\geq$ CO threshold (2.2x1018 for NH and 1.8x1018 for SH) or when CO values larger than 2.8x1018 (2.5x1018) are observed in the Northern (Southern) Hemisphere regardless of AI considerations. Conversely, OMI pixels with observed AI $\geq$ AI threshold (0.8) and CO <CO threshold are assigned with the dust aerosol type. Threshold values in AI and CO represent noise and background levels in the respective measurements not necessarily associated with the free troposphere CO burden which is expected to co-exist with the lofted carbonaceous aerosols.

The straightforward way of discerning the absorbing aerosol type works efficiently in most cases, however, may break down under certain situations, i.e., when dust aerosols are present over regions characterized by high CO levels associated with pollution episodes other than the biomass burning smoke for which the scheme would assign absorbing aerosol type as smoke. Note that the aerosol type identification scheme doesn't account for the mixture of aerosols for which either smoke or dust aerosol type is assigned depending upon the threshold values of AI and CO.

A detailed regional-level comparison between CALIOP aerosol sub-type and that of the

OMAERUV hasn't been done, but we consider conducting the said analysis in the near future. A brief description of the aerosol type identification scheme is now provided in the revised paper in section 2.2.2.1.

RC: Single scattering albedo (section 2.2.2.1): the SSA is extremely important for ACA retrieval and DRE. I hope the information of aerosol type and the corresponding SSA will be part of the OMIACA so the users can use it in a consistent way with the AOD product

AR: We fully agree with the referee that the assumption of aerosol SSA above the clouds is extremely crucial for deriving accurate ACAOD retrievals as well as in the quantification of DRE. Section 2.2.2.1 describes how we take advantage of clear-sky SSA retrievals from the OMAERUV product and assign a representative SSA value for each region at a daily scale. In this regard, the OMACA stands alone among passive-sensor based ACA algorithms that currently rely on a fixed value of SSA [Jethva et al., 2013; Myer et al., 2015].

Looking at its importance, the values of aerosol SSA above-cloud assumed in the OMACA algorithm for the three wavelengths, i.e., 354, 388, and 500 nm, are already reported in the product. The corresponding SDSs are named as "InputSSA354", "InputSSA388", "InputSSA500". Additionally, the aerosol type associated with each valid ACA pixel is also stored in the product as "AerosolType". Refer to the complete list of SDS stored in each OMACA HDF-EOS file in Appendix I.

RC: Look-up-tables (section 2.2.24): How is cloud effective radius considered in the LUT? Is it assumed as a constant? Note that the assumption of CER could have impacts on the COD retrieval. Some discussion is needed to clarify this.

AR: In LUT calculations, clouds are assumed to be liquid in phase and follow the standard C1 size distribution [Deirmendjian, 1969]. The effective radius of C1 water cloud droplet distribution is assumed to be a constant value of 6.0 microns. To answer the reviewer's concern, we carried out a sensitivity analysis, similar to the ones presented

in Table 3, 4, and 5, in which the errors in both ACAOD and aerosol-corrected COD were calculated following the perturbation approach around the assumed CRE value of 12.0 $\mu$m. The table attached to this response lists the errors in aerosol-corrected COD due to the range of uncertainty in the assumed cloud CRE. The analysis was performed assuming reference cloud CRE of 12 $\mu$m and for the ACAOD values of 0.5 and 1.0 (388 nm).

The errors in COD retrievals due to the uncertainty in CRE follow asymmetric behavior to the perturbation around the assumed state. While a large underestimation in CRE of -8 $\mu$m produces negative errors of $\sim$10%-11% in the retrieved COD, an overestimation in CRE of +8 to +12 $\mu$m yields positive errors of much smaller magnitudes ($\sim$1%-2%). The spatial distribution of MODIS monthly cloud CRE over the Southeastern Atlantic Ocean, as shown in Figure 11 of Meyer et al. (2015), exhibits spatial variations with smaller droplets (CRE 7-11 $\mu$m) concentrated closer to the coast and relatively larger size droplets (11-17 $\mu$m) retrieved away from the coast. Given the fixed value of CRE equals 6.0 $\mu$m assumed in the OMACA cloud LUTs, the observed variations from MODIS would produce <2% error in the retrieved aerosol-corrected COD.

The corresponding errors in ACAOD due to the uncertainty in cloud CRE are found to be marginal. For an ACAOD>0.5, an uncertain assumption in cloud CRE by $\pm$8 $\mu$m results in ACAOD errors <2% with much smaller magnitudes at higher aerosol loading. This is because at larger ACAODs the aerosol absorption effects dominate over that resulting from varying effective radius of liquid droplets leaving other major algorithmic assumptions, i.e., SSA, ALH, and AAE to determine the resultant uncertainty in ACAOD retrievals.

This analysis implies that near-UV wavelengths don't offer a strong sensitivity to the variations in cloud droplet size rather the cloud signal is predominantly driven by the optical thickness. Due to the lack of information on cloud droplet size from OMI, we adopted the standard C1 cloud model validated and used in numerous studies for all cloud LUT calculations.

The description provided above is now added to the section 3 of the revised paper.

RC: Partly cloudy pixels: the footprint size of OMI is 13x24 km (338km2). At this scale, there cloud be a lot of partly cloudy pixels. One of my main questions/concerns is about the treatment of the partly cloudy OMI pixels. It seems to me the OMIACA algorithm is applied to both overcast pixels and partly cloudy pixels, correct? How is the subpixel cloud fraction determined? How are the partly cloudy pixels treated in the LUT and radiative transfer simulations? How is the UVAI of clear-sky part of the partly cloudy pixel different from that of the cloudy part, and what is the meaning of the "observed" UVAI for the partly cloudy pixel? I would strongly recommend the authors to add a separate and dedicated sub-section to discuss the treatment of partly cloudy pixels in the OMIACA algorithm.

AR: OMACA algorithm performs retrievals for each pixel of size 13 x 24 km-square at nadir independently. As the referee has correctly pointed out, there is a possibility of encountering partly cloudy pixels, especially for measurements with lower reflectivity (388 nm) values. The algorithm quality flags reported in OMACA product precisely reflect these observed conditions (Table # 1 QFlag values 0, 1, and 2). Due to the coarser resolution of OMI pixels, there seem to be is no direct way to infer the sub-pixel cloud variability using only OMI measurements. Therefore, we have used the OMI-MODIS joint cloud product, OMMYDCLD, post-retrievals for all analyses reported in the manuscript. The OMMYDCLD product, as already explained in the original manuscript in section 2.2.3, provides the statistics of the MODIS 1-km cloud product (MYD06) on each collocated OMI footprint.

An analysis using the OMMYCLD product over the Southeastern Atlantic Ocean for the period of Jun-July-Aug 2007 revealed a well-constrained non-linear relationship between the MODIS-derived COD times the geometric cloud fraction and LER388. A threshold of LER388 of 0.25 adopted for the best quality retrievals (QFlag=0) compares to the COD times geometric cloud fraction of 3-4. Conversely, given the geometric cloud fraction of unity, the minimum COD retrieved by OMACA would be in the range 3-

4. Retrievals assigned with QFlag=1 further extends the LER388 to 0.20 allowing pixels with relatively lower reflectivity with much stronger absorption (larger UVAI) above the clouds [Jethva et al., 2013, Figure 6, Aug 12, 2006 case study].

Currently, the OMACA algorithm is designed to perform inversion over fully overcast pixels. The LUTs are generated assuming fully cloudy conditions and do not explicitly treat partly cloud pixels. To avoid a large fraction of partly cloudy pixels, therefore, we adopted the geometric cloud fraction thresholds calculated using OMMYDCLD product of 0.5 and 0.75 for the FOACA (Figure 5, 6, 7) and AOD/COD analyses (Figure 8, 9, 10, 11).

We will consider using the OMMYDCLD product in the OMACA processing in the next upgrade of the algorithm. We believe that most of the explanation provided here on this concern is already described in the original version of the paper in section 2.2.3 and 2.2.4.

RC: Sub-pixel COD variation: A related question is whether and how the algorithm accounts for the subpixel COD variation. What is the physical meaning of the "retrieved COD"? Is it a simple mathematical mean or some kind of weighted mean?

AR: As explained in the previous response, each OMACA retrieval corresponds to the respective pixel size derived using a single set of reflectivity and UVAI values. The retrieved values of ACAOD and COD, therefore, represent an overall condition observed in each pixel. It is hard to draw a conclusion that the observed condition is merely a mathematical mean or weighted mean of the sub-pixel cloud variability as the TOA reflectance versus COD relationship exhibits a non-linear behavior especially at higher values of CODs.

RC: Spatial distribution of ACA: As mentioned above Zhang et al. (2016) studied the global distribution of different types of ACA. Actually, Figure 5 agrees reasonably well with the Figure 2 of Zhang et al. (2016). In addition, Zhang et al. (2016) also found significant amount of ACA over the north pacific due to the Asian dust and pollution.

This study should be cited here anda discussed.

AR: We referred to Zhang et al. [2016] paper which also shows similar cloudy-sky FOACA results over the global ocean derived using 8-year daytime CALIOP observations. Prior to Zhang et al. [2016] and our study, Devasthale and Thomas (2011) also conducted FOACA analysis using 4-years of CALIOP data. Both studies are now mentioned and cited in the revised manuscript.

RC: Figure 5 shows some ACA over the Southern Ocean in January and February. Is this true or some retrieval artifact?

AR: Most likely ACA over the Southern Ocean during the winter months inferred from our analysis is an artifact resulting from non-aerosol related enhancement in UVAI observed at certain geometry conditions that are associated with higher solar zenith and viewing zenith angles. Although these artifacts are largely removed from the best quality retrieval group (see Table 1 of the original manuscript) based on thresholds in geometry and are assigned a different quality flag (=3), some residual ACA pixels still reside within the group of good quality retrievals. One of the possible reasons for the non-aerosol related enhancement in the UVAI could be the presence of ice clouds over the Southern Ocean for which the effect of angular scattering is unaccounted for in the calculation of UVAI.

RC: It is interesting to see that the FOACA in Figure 5 and the ACA AOD in Figure 8 are highly correlated. Is this a coincidence or there may be some real connection between them? One could argue that the region with high FOACA does not necessarily have a larger ACA AOD. Do you agree? AR: It is assumed here that the reviewer is referring to the high-level of spatial consistency in FOACA and ACAOD maps. This is because of the OMACA algorithm's efficiency to quantitatively retrieve ACAOD and COD when the presence of absorbing aerosols over clouds is identified based on a set of LER and UVAI thresholds. We agree with the reviewer that high-frequency occurrence and large ACAOD do not necessarily correlate with each other. FOACA is simply a measure of

the temporal occurrence of the aerosol-cloud overlap, whereas ACAOD is a quantitative measure of actual aerosol loading above the clouds.

[Figure]

**Theoretical error (%) in aerosol-corrected COD (388 nm) due to the uncertainty in the assumption of cloud effective radius**

| Error in cloud effective radius (µm) | Assumed AOD (388 nm) = 0.5/1.0 Reference cloud effective radius = 12.0 µm % Error in Aerosol-corrected COD (388 nm) | | | |
|---|---|---|---|---|
| | COD=5 | COD=10 | COD=20 | COD=30 |
| -8.0 | -11.31/-11.9 | -10.28/-10.88 | -9.13/-9.83 | -10.11/-10.76 |
| -6.0 | -5.94/-6.54 | -5.38/-5.94 | -4.68/-5.29 | -5.01/-5.62 |
| -4.0 | -2.85/-3.26 | -2.57/-2.94 | -2.18/-2.57 | -2.25/-2.66 |
| 4.0 | 1.17/1.4 | 1.43/1.79 | 1.21/1.62 | 0.95/1.39 |
| 6.0 | 1.34/1.70 | 1.63/2.07 | 1.37/1.87 | 1.05/1.59 |
| 8.0 | 1.51/1.94 | 1.83/2.35 | 1.52/2.11 | 1.16/1.78 |
| 12.0 | 1.85/2.41 | 2.22/2.92 | 1.83/2.59 | 1.36/2.18 |

**Fig. 1.**